# Spontaneous cortical activity transiently organises into frequency specific phase-coupling networks

Diego Vidaurre[1,2,3,4], Laurence T. Hunt [1,2,3], Andrew J. Quinn[1,3], Benjamin A. E. Hunt[5,6], Matthew J. Brookes[5], Anna C. Nobre[1,3,7] & Mark W. Woolrich[1,2,3]

Frequency-specific oscillations and phase-coupling of neuronal populations are essential mechanisms for the coordination of activity between brain areas during cognitive tasks. Therefore, the ongoing activity ascribed to the different functional brain networks should also be able to reorganise and coordinate via similar mechanisms. We develop a novel method for identifying large-scale phase-coupled network dynamics and show that resting networks in magnetoencephalography are well characterised by visits to short-lived transient brain states, with spatially distinct patterns of oscillatory power and coherence in specific frequency bands. Brain states are identified for sensory, motor networks and higher-order cognitive networks. The cognitive networks include a posterior alpha (8–12 Hz) and an anterior delta/theta range (1–7 Hz) network, both exhibiting high power and coherence in areas that correspond to posterior and anterior subdivisions of the default mode network. Our results show that large-scale cortical phase-coupling networks have characteristic signatures in very specific frequency bands, possibly reflecting functional specialisation at different intrinsic timescales.

[1] Wellcome Centre for Integrative Neuroimaging, Oxford Centre for Human Brain Activity (OHBA), University of Oxford, Oxford OX37XJ, UK. [2] Wellcome Centre for Integrative Neuroimaging, Oxford University Centre for Functional MRI of the Brain (FMRIB), University of Oxford, Oxford OX39DU, UK. [3] Department of Psychiatry, University of Oxford, Oxford OX37JX, UK. [4] Department of Brain Physiology, Graduate School of Frontier Biosciences, Osaka University, Osaka 565-0871, Japan. [5] School of Physics and Astronomy, University of Nottingham, Nottingham NG72RD, UK. [6] Department of Diagnostic Imaging, The Hospital for Sick Children, Toronto, ON M5G 1X8, Canada. [7] Department of Experimental Psychology, University of Oxford, Oxford 0X26GG, UK. Correspondence and requests for materials should be addressed to D.V. (email: diego.vidaurre@ohba.ox.ac.uk)

Efficient neuronal coordination between regions across the entire brain is necessary for cognition[1–4]. A proposed mechanism for such coordination is oscillatory synchronisation, that is, populations of neurons transmit information by coordinating their oscillatory activity with the oscillations of the receptor population at certain frequencies. Furthermore, different frequencies, or, more generally, different oscillatory patterns, subserve different functions[5]. At the same time, phase-coupling between neuronal populations in specific frequency bands has been proposed as a mechanism for regulating the integration and flow of cognitive content[6–8] and coordinating neuronal spike timing[9]. The role of phase-coupling at distinct frequencies has also been demonstrated in tasks at the large scale, where task-relevant information is effectively transmitted through phase-locking between separate cortical regions[6,10–12].

Using functional magnetic resonance imaging (fMRI), it has been shown that large-scale networks activated in tasks are also spontaneously recruited in the resting state[13]. These networks have previously been shown to have distinct band-limited power in electroencephalography (EEG) and magnetoencephalography (MEG)[14–18]. If these spontaneously occurring networks are to provide an effective substrate for cognitive processes, then they might also be expected to exhibit the same fast changing phase-coupling activity observed in tasks[6,19,20]. However, the evidence for frequency-specific phase-coupling in spontaneous activity at timescales associated with fast cognition is limited.

Here we propose that cortical activity at rest can be described by transient, intermittently reoccurring events in which large-scale networks activate with distinct spectral features that include both power and phase-coupling. To identify the possible presence of these events, we use a new analysis approach based on the Hidden Markov Model (HMM)[21], a general mathematical framework previously used to find recurring states in brain data[22]. For the first time, this allows for the identification of brain-wide networks (or brain states) characterised by specific patterns of power and phase-coupling connectivity, which, crucially, are spectrally resolved (i.e. power and phase-coupling are defined as a function of frequency). These patterns are also temporally resolved, meaning that the method provides a probabilistic estimation of when the different networks are active (see Fig. 1a). We used resting-state MEG data from 55 healthy subjects, source-reconstructed to 42 regions across the entire cortex. Notably, applying this approach to these data revealed the distinct temporal and spectral properties of anterior versus posterior regions of the default mode network (DMN). The joint description of the spectral, temporal and spatial properties of ongoing neuronal activity provides new insight into the large-scale circuit organisation of the brain[23].

## Results

**Twelve states were identified using HMM.** Using concatenated MEG resting-state data from 55 subjects, mapped to a 42-region parcellation using beamforming[24] with reduction of spatial leakage in order to diminish the effects of volume conduction[25], we identified 12 HMM states using a novel approach that we refer to as time-delay embedded HMM (TDE-HMM). Essentially, this technique finds, in a completely data-driven way, recurrent patterns of network (or HMM state) activity. Each HMM state has parameters describing brain activity in terms of power covariations and, crucially, coherence between every pair of regions. The method provides information that is both spectrally (power and phase-coupling are defined as a function of frequency) and temporally resolved (different networks are described as being active or inactive at different points in time). Importantly, while the spatial and spectral description of the states is common to all subjects, each subject has their own state time course, representing the probability of each HMM state being active at each instant (see Methods for further details). See Fig. 1a for a graphical example, and Supplementary Fig. 1 for an illustration of the entire pipeline, and refer to Methods for links to the code repository and examples.

**The states exhibit specific phase-locking connectivity.** Figure 2 shows spatial maps of power and phase-coupling connectivity, both averaged across a wideband frequency range (1–45 Hz), for 4 of the 12 estimated states. The power maps are in relation to the mean power across states, and all maps are thresholded for ease of visualisation; see Supplementary Fig. 3 for a statistical testing analysis on power and connectivity and Methods for further details. Supplementary Fig. 2 shows the remaining eight states, four of which exhibit reduced power and connectivity relative to

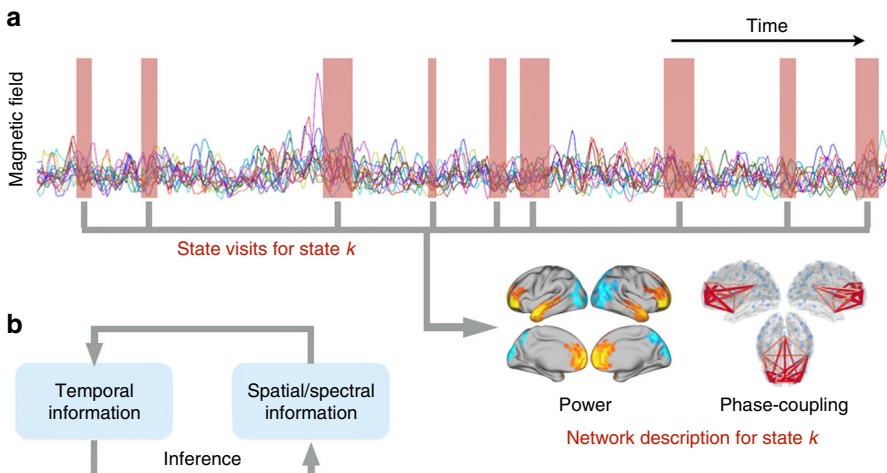

**Fig. 1** Schematic illustration of the method. **a** Each state is defined as having its own distinct temporal, spatial and spectral characteristics. The temporal information is given by when the state is active (red boxes). The spatial and spectral descriptions of the power maps and phase-coupling networks are contained in the parameters of each state. **b** Schematic of the iterative model inference. The state parameters are estimated using those segments of the data for which the state is currently estimated to be active. In turn, the estimation of when a state is active is based on the how well each state can explain each time point (i.e. according to the current estimate of the spatial/spectral state parameters)

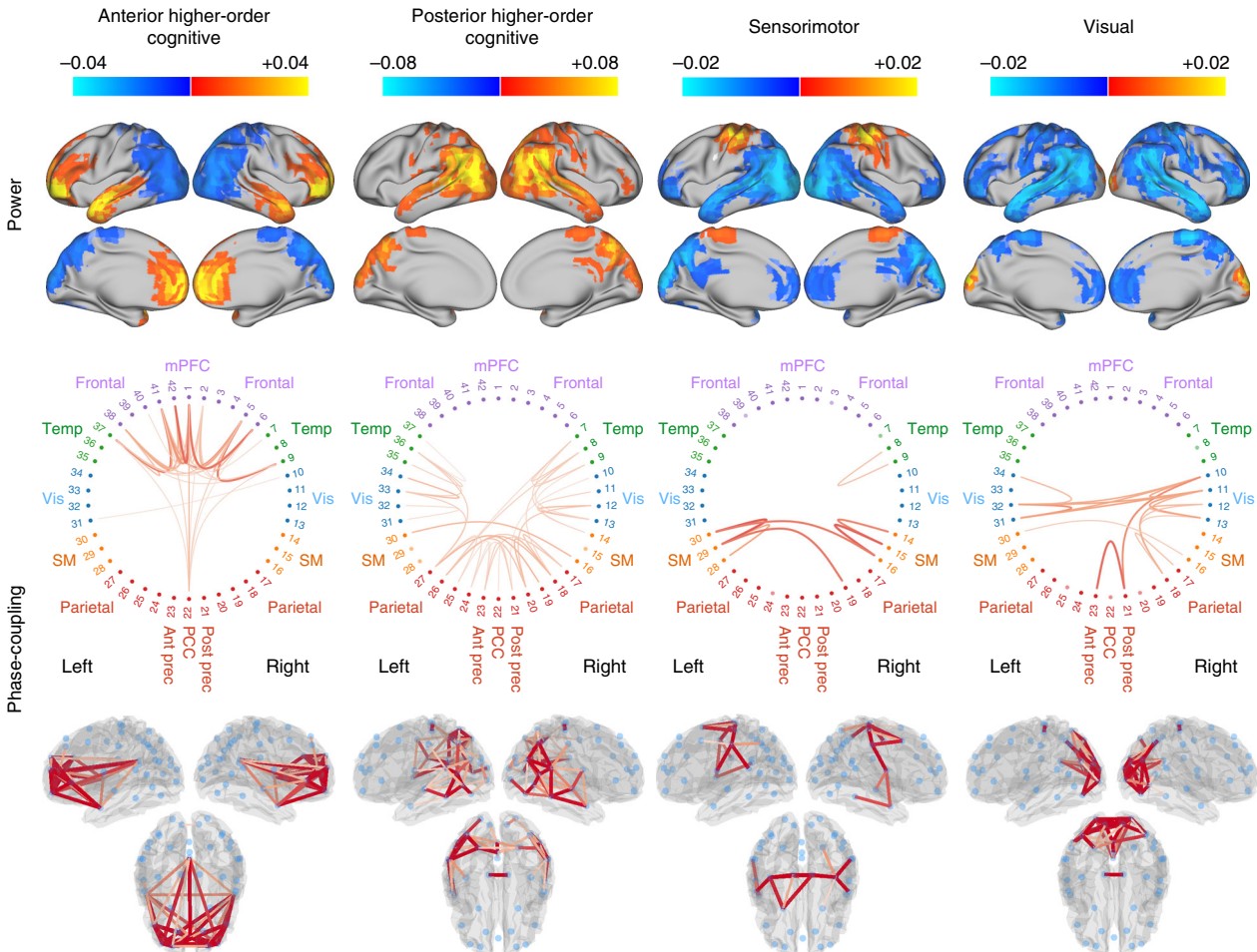

**Fig. 2** Brain states identified using Hidden Markov Modelling represent networks of spectral coherence. Wideband (1–30 Hz) thresholded power maps and phase-coupling are displayed for the two higher-order cognitive (anterior and posterior) states and the visual and motor states. The two higher-order cognitive networks contain regions that suggest a subdivision of the default mode network. Power maps are relative to the temporal average, i.e. they are globally centred such that blue colours reflect power that is lower than the average over states and red/yellow colours reflect power that is higher than the average over states. The coherence networks only show high-valued connections (see Methods). In the circular phase-coupling plots, each numbered dot represents one brain region. Supplementary Fig. 1 shows the same information for the other eight states

the grand average. The power maps and phase-coupling connectivity of each state tend to be (although not exclusively) bilateral, with strong increases in power tending to (although not exclusively) accompany increases in phase-locking. We refer to two of the states (left) as being "higher-order cognitive", in accordance with the brain areas they incorporate and previous literature[13,26–28]. The other two states (right) correspond well to visual and motor systems. The two higher-order cognitive networks involve regions that together form the DMN; see below. This affords the interpretation that the DMN, when analysed at the finer timescales, can be decoupled into two separate components. The anterior higher-order cognitive state includes the temporal poles (often associated with semantic integration[29]) and the ventromedial prefrontal cortex (typically implicated in emotion regulation and decision making[30]), exhibiting a strong connectivity with the posterior cingulate cortex (PCC), which is a key region of the DMN[31,32]. The posterior higher-order cognitive network encompasses the PCC/precuneus, bilateral superior and inferior parietal lobules; bilateral intraparietal sulci; bilateral angular and supramarginal gyri and bilateral temporal cortex. These regions are classically associated with integration of sensory information, perceptual-motor coordination and visual attention, as well as processing of sounds, biological motion and theory of mind[33].

**Spectral features of the higher-order cognitive states.** Previous work looking at the global (temporally averaged) estimates of large-scale functional connectivity has demonstrated that different brain networks show correlation of power in different frequency bands[34]. Leveraging the fact that our model is spectrally resolved, we sought to investigate how power and phase-coupling varies with frequency in the different brain states.

For the four states shown in Fig. 2, Fig. 3a shows power versus coherence, with dots representing each brain region. These results are shown wideband (1–45 Hz) and for three different frequency modes. The frequency modes were estimated following a data-driven approach (non-negative matrix factorisation, see Methods), which identified frequency modes that approximately correspond to classical frequency bands (although overlap one another to a certain extent, bringing some data-driven flexibility). For convenience, we labelled the data-driven modes using the closest corresponding classical frequency bands, resulting in "delta/theta" (0.5–10 Hz), "alpha" (5–15 Hz), "beta" (15–30 Hz) and "low gamma bands" (30–45 Hz). It should, however, be kept in mind that the frequency modes are derived from the data and so are not exactly the same as the classical frequency bands normally used. Possibly owing to the relatively low signal-to-noise ratio in higher frequency bands, strong state-specific differences in the gamma band could not be observed with this approach,

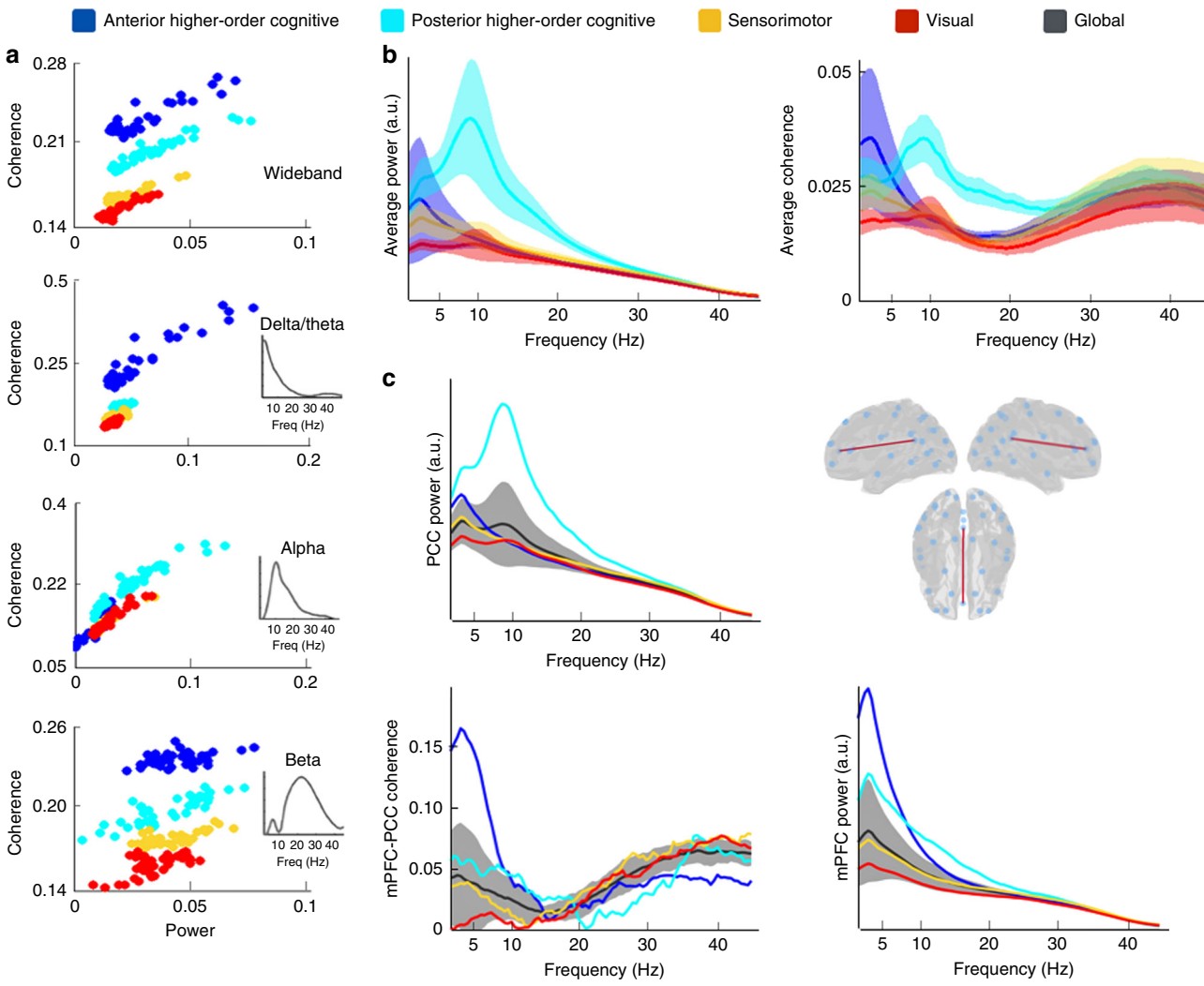

**Fig. 3** The higher-order cognitive states have distinct spectral features compared with the other states. **a** Total connectivity of each region (defined as the sum of the values of coherence of the region with the rest of the regions) against power, for wideband and the three estimated frequency modes (see Methods), where each dot represent a different brain region. Both power and connectivity are higher for the higher-order cognitive than for the visual and motor states, with coherence exhibiting the largest difference. **b** Spectral profiles of the two higher-order cognitive (anterior and posterior) and the visual and motor states, in terms of power averaged across brain regions (left) and coherence averaged across all pairs of brain regions (right); shaded areas represent the standard deviation across brain regions (or pairs of regions). Supplementary Fig. 5 shows the power spectra for the anterior/posterior precuneus alongside the PCCs. **c** Power for PCC (top left), power for mPFC (bottom right) and coherence between mPFC and PCC (bottom left) for the four considered states in comparison to the grand average (black line, with the shaded areas representing standard deviation across states). The temporally average global power and coherence has a relative lack of spectral detail compared with the individual brain states

and therefore, we only show results for the delta/theta, alpha and beta modes. Strong increases in power tended to (although not exclusively) accompany increases in coherence. Interestingly, the differences in coherence between the states are much more pronounced than the differences in power.

To expand on the specific spectral differences between the higher-order cognitive and the visual and motor states, Fig. 3b shows power and coherence averaged across all brain regions as a function of frequency. This shows frequency at full spectral resolution rather than the frequency modes used in the previous Fig. 3a. The anterior higher-order cognitive state is characterised by strong power and coherence in the slowest frequencies (delta/theta), whereas the posterior higher-order cognitive is dominated by the alpha frequency. More specifically, there is a strong component at ~4 Hz for the anterior higher-order cognitive state in both power and coherence and a component at 10 Hz for the posterior higher-order cognitive state also in both power and

coherence. These results reveal that the two higher-order cognitive states, which may correspond to subdivisions of the DMN, exhibit more power and coherence than the visual and motor states. Moreover, they have very different dominant frequencies.

Given the key role attributed to the PCC and the medial prefrontal cortex (mPFC) within the DMN and resting-state networks more broadly[31,32,35], we next examined the state-specific frequency profile of the PCC and the mPFC to see if their spectral characteristics in the higher-order cognitive states are significantly different to the other (less cognitive) states. Figure 3c shows the power in the PCC and the mPFC, as well as the coherence between these two regions. The four considered states were compared to the global average (solid black lines; shaded areas represent the standard deviation across states), which corresponds to the power and coherence computed from a static (rather than dynamic) perspective. The PCC has more power

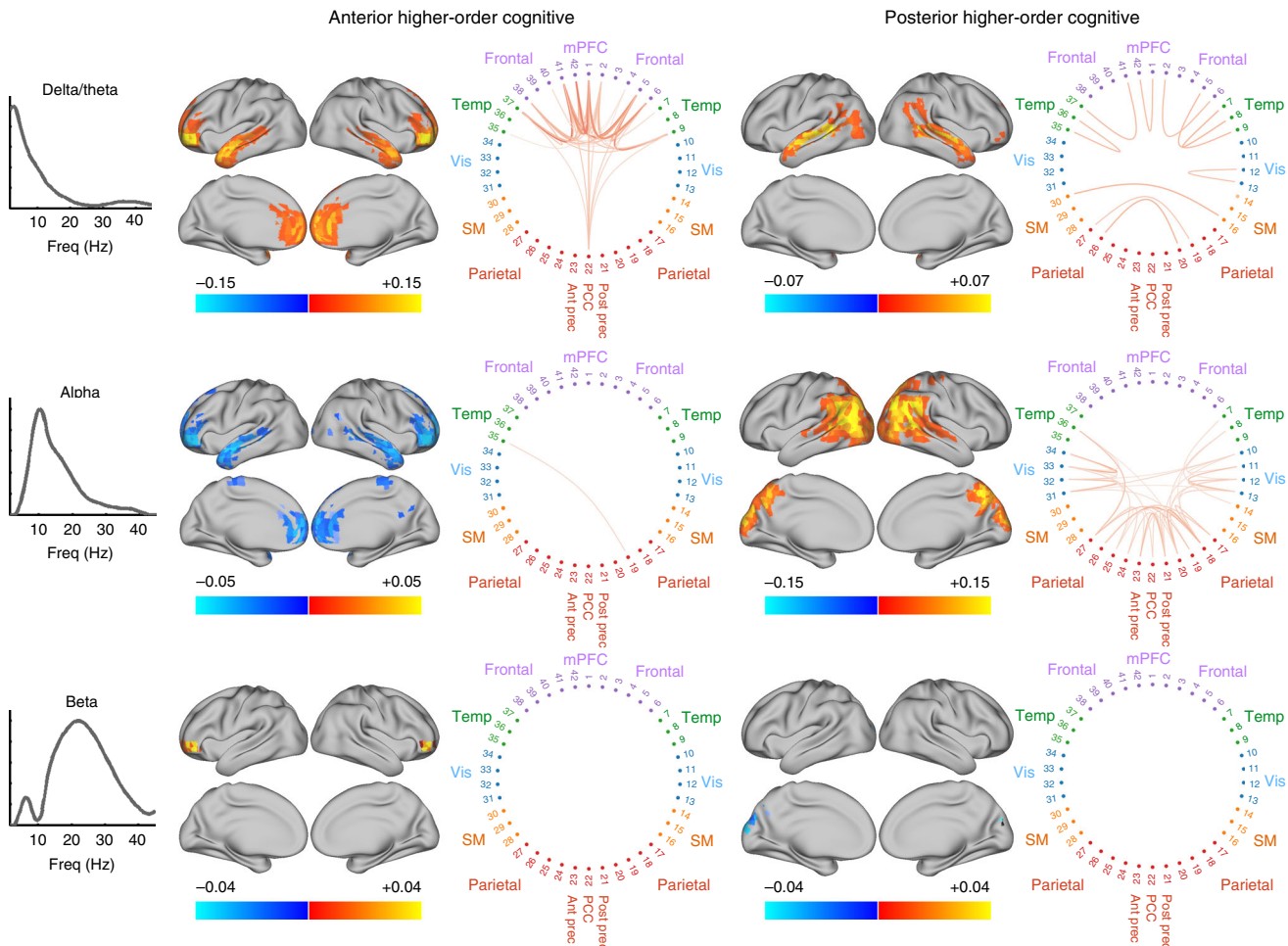

**Fig. 4** The two higher-order cognitive states operate in different frequencies. Frequency-specific relative power maps and phase-coupling (see Fig. 2 for details) for the anterior and posterior higher-order cognitive states and for the three data-driven estimated frequency modes. Whereas activity (power and phase-locking) is dominant in the delta/theta frequencies for the anterior-cognitive state, the posterior-cognitive state is dominated by alpha. Both states exhibit strong phase-coupling with the PCC but in different frequency bands. Supplementary Fig. 4 shows a similar view of the visual and motor states

across all frequencies in the posterior higher-order cognitive state, although the power in the slow frequencies for the anterior higher-order cognitive state is also significantly above the global average. By contrast, the mPFC shows high power in the anterior higher-order cognitive state, particularly in the delta/theta frequency range. Finally, global phase-coupling is high in the anterior higher-order cognitive state in delta/theta, whereas the posterior higher-order cognitive state exhibits high PCC connectivity in alpha and beta. Altogether, these results suggest that (i) the PCC has spectral properties that are unique to the higher-order cognitive states (consistent with the idea of the PCC being a hub region), (ii) the anterior higher-order cognitive state involves the PCC in slower frequencies than the posterior higher-order cognitive state for both power and phase-coupling connectivity, and (iii) these properties are only observed when we compute power and coherence specifically within the fast transient events that correspond to the HMM brain states, whereas the global, temporally averaged properties of the PCC (black line) are far less striking.

We next examined the spatial distribution of these spectral differences using the frequency modes identified in Fig. 3. Figure 4 shows power and phase-coupling in brain space for each cognitive state and frequency mode (Supplementary Fig. 4 presents a similar view for the visual and motor states). This view clearly reflects that the two higher-order cognitive states

have a distinct spatial distribution of power and connectivity. For example, we observe strong phase-coupling between frontal areas, mPFC and the PCC specifically in the delta/theta mode for the anterior higher-order cognitive state. By contrast, the posterior higher-order cognitive state is characterised by pronounced intraparietal and PCC/precuneus connectivity (specifically in the alpha frequency mode) and by some slow frequency power and phase-coupling in the temporal regions. As observed in Fig. 2, both power and connectivity exhibit strong interhemispheric symmetry.

**Temporal features of the higher-order cognitive states**. Together with the state distributions, the HMM inference also estimates the time courses of the visits to each of the brain states. We used these to look at the extent to which the temporal characteristics of the higher-order cognitive states differed to the visual and motor states. Figure 5a shows for each state: the dwell times (or life-times, i.e. the amount of time spent in a state before moving into a new state), the interval times between consecutive visits to a state, and the fractional occupancies (reflecting the proportion of time spent in each state). All HMM states were on average short-lived, their dwell times lasted on average between 50 and 100 ms. Note that, as shown empirically in the Supplementary Note 1, coherence can still be reliably measured for short

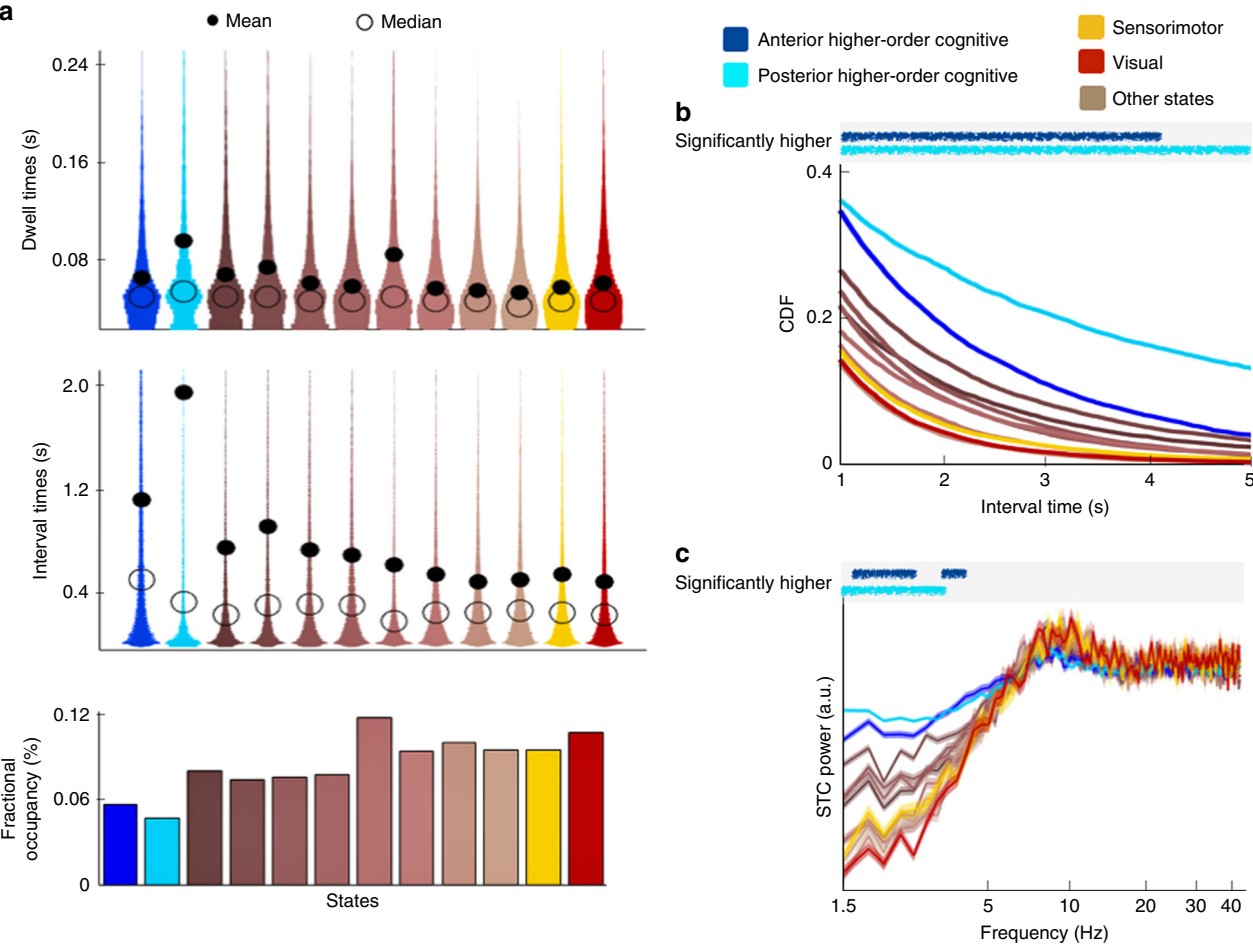

**Fig. 5** The higher-order cognitive states have distinct temporal features compared with the other states. The states depicted in brown colours are depicted in Supplementary Fig. 2 in an order that consistent with this figure (i.e. the first four are states with positive activation and the second four have a negative activation). **a** Distribution of state dwell times (time spent in each state visit), distribution of interval times between state visits and fractional occupancies (proportion of time spent in each state). The dwell times are significantly longer for the posterior higher-order cognitive state than for all the other non-cognitive states ($p$ value < 0.001), and the interval times are significantly longer for the two higher-order cognitive states than for the other states ($p$ value < 0.001). **b** Cumulative density function (CDF) for the interval times, reflecting much larger tails for the interval time distribution of the two higher-order cognitive states, both of which have significantly larger CDF values than the other states (permutation testing; statistical significance for a confidence level of 0.01 is indicated by the lines on top of the panel). **c** Spectral analysis of a point process representing the onset of the state events, computed separately for each state (99% confidence intervals are indicated by shaded areas); no slow oscillatory modes in the state occurrences themselves is revealed. The higher-order cognitive states have a stronger power in the 1.5–5 Hz range of frequency than the rest of the states (statistical significance using permutation testing is indicated on top, using a confidence level of 0.01)

state visits even at the slowest frequencies. Since the absolute value of these temporal features must be interpreted with caution given the state exclusivity assumption of the HMM (see Discussion for details), we examined only relative differences between the states.

We observed longer dwell times for the posterior higher-order cognitive state than for the states that were not higher-order cognitive states (permutation testing, $p$ value < 0.001). However, the largest differences are found in the interval times. Both of the anterior and posterior higher-order cognitive states have visits that are much more temporally separated than the other states (permutation testing: $p$ value < 0.001 for both tests). The interval time distributions of the posterior higher-order cognitive state and, to a lesser extent, of the anterior higher-order cognitive state, have pronounced tails for higher interval times, as indicated by the mean of the distribution being much larger than the median. To further illustrate this, Fig. 5b shows the cumulative density function (CDF) of the interval times, which evaluates the proportion of intervals ($y$ axis) that are longer than any given

interval duration ($x$ axis). The CDF is particularly useful to examine the differences between the tails of the distributions. We observe that both higher-order cognitive states have significantly larger CDF values than the other states (significance for a confidence level of 0.01 is indicated by the lines on top of the panel, using permutation testing). For example, the time between state visits is >1 s, in ~40% of the higher-order cognitive state visits, as compared to only ~20% of the time for the other states. Importantly, this is not due to differences in fractional occupancy (depicted in the bottom panel of Fig. 5a), given that the fractional occupancies of the higher-order cognitive states are not significantly different from the visual and motor states. In summary, these results indicate that the higher-order cognitive states tend to last longer, but are not revisited for longer periods, than the visual and motor states.

A related point is the extent of state fractional occupancy variability in the across subjects and whether this distribution is relatively flat or varies strongly across subjects. Supplementary Fig. 6 reveals that the distribution is indeed not uniform and that

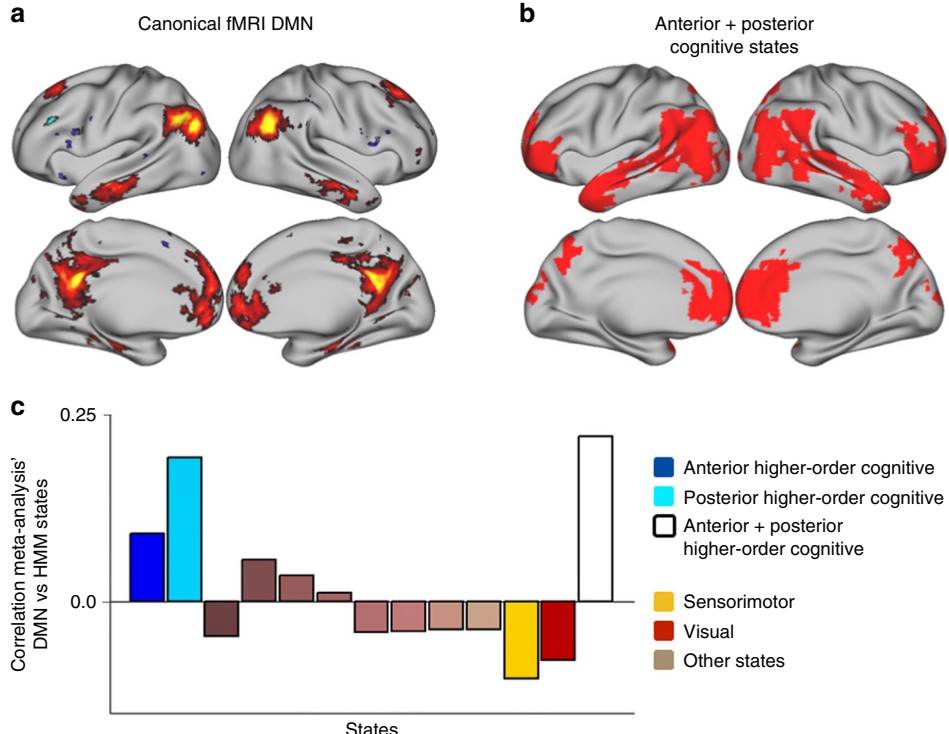

**Fig. 6** The higher-order cognitive states correlate with the canonical fMRI DMN, according to a meta-analysis. **a** Canonical DMN derived from the fMRI literature. **b** Union of the two higher-order cognitive states (see the corresponding section in Results). **c** Correlation of each state to the canonical fMRI DMN; note that the far right bar, shown in white, corresponds to the union of the two higher-order cognitive states shown in **b**

different subjects have different degrees of state representation, which might possibly relate to specific subject traits[36].

We also investigated whether brain states are visited rhythmically, i.e. aligned to the peaks and troughs of a possible oscillation. Modelling the state events as a point process (where a state occurrence is defined at the onset of the state visit), we computed the spectra of these processes to find whether there were any strongly characteristic frequencies. Figure 5c shows the power of the event point process for all states (shaded areas reflect 99% confidence intervals), which does not show any strong frequency mode and has particularly low power in the slow frequencies. This is in line with recent findings in task[37], where it is only through trial-averaging (given the temporal variability of the task-related events within each trial) that these events appear as sustained oscillations. Also, consistently with the above results, we observed a higher power in between 1 and 5 Hz for the higher-order cognitive states (significance for a confidence level of 0.01, using permutation testing, is indicated on top), reflecting the longer dwell and interval times for these states.

**The higher-cognitive states and the canonical fMRI DMN.** Throughout the paper, we have drawn a link between the two identified higher-cognitive states and the DMN. We now evaluate quantitatively this relation using a meta-analysis technique. In particular, using Neurosynth[38], we extracted a canonical map of DMN activation from the fMRI research literature (Fig. 6a). Using the wideband power maps (as shown in Fig. 2), we computed masks by selecting the 10% most active voxels for each state. We then defined the union of the two higher-cognitive states (depicted in Fig. 6b) by selecting the voxels that are active in either of the two corresponding masks. Using these masks, Fig. 6c shows the correlation of each state to the canonical DMN map. As observed, the two higher-order cognitive states hold the highest correlation of all states with the canonical DMN, with

the union of the two higher-cognitive states being more correlated to the canonical DMN map than any individual state. This result quantitatively demonstrates the relation of the two higher-cognitive states to what is understood by the canonical DMN as measured by fMRI.

**Power versus connectivity in driving state switching.** An important question is which features in the data (i.e. power or connectivity) are driving the HMM state segmentation. Given that power can be estimated more precisely than cross-spectral properties (such as coherence) and that the leakage correction procedure[25] may remove some genuine connectivity information (if interactions occur with, or close to, zero-lag), changes in power are expected to drive a considerable amount of state switching. By manipulating the diagonal and off-diagonal elements of the autocovariance matrices that characterise each state, we computed the Riemannian distance (see Methods) between each pair of states using (i) power and coherence, (ii) just coherence and (iii) just power. Figure 7a reflects that the contribution of power and coherence to state differences is variable, but, on average, the contribution of power is indeed around four times higher on average (note the difference in the scale bar).

However, this does not mean that phase coupling does not contribute to the inference. To demonstrate this, we ran the HMM on power envelopes computed from the data band-passed filtered between 1 and 40 Hz[39], which defines HMM states as having distinct patterns of power and power correlations in a single frequency band. We then computed the spatial correlation between the power maps and the coherence connectivity profiles (in either case using the multitaper after the HMM inference) between the two types of HMM. We paired the states between the two runs such that the correlations are maximal. Figure 7b shows that some power maps are relatively well correlated between the two runs and that the differences of functional connectivity

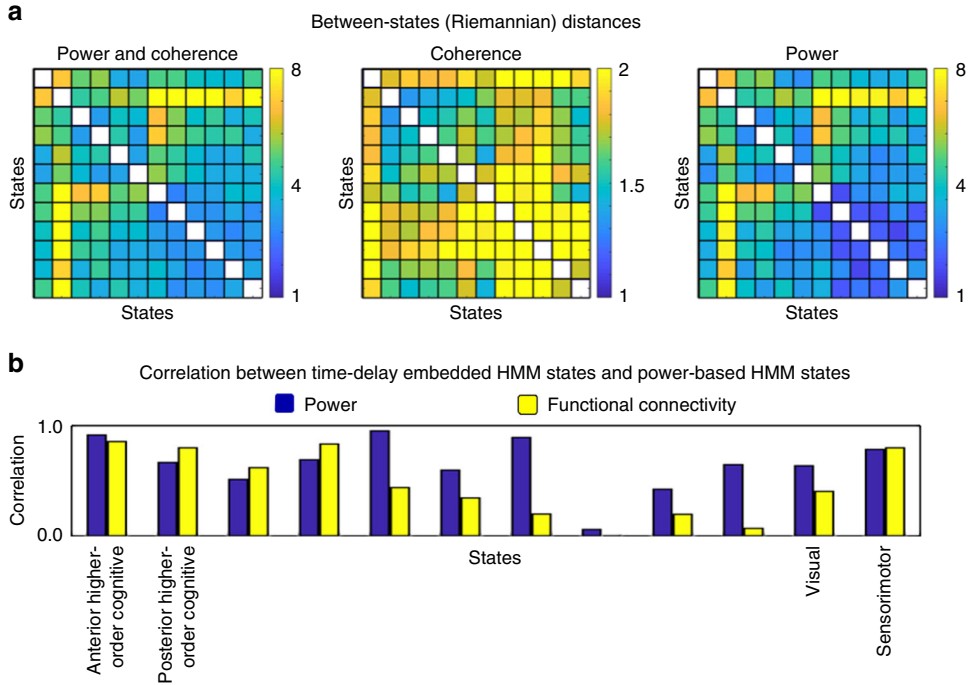

**Fig. 7** Oscillatory power contributes more than phase-coupling to state switching. **a** Riemannian distances between states when we use power and coherence, only coherence and only power. **b** Correlation between the TDE-HMM and power-based HMM states run on the same data

between the two runs are in general larger than the differences in power. In summary, this analysis demonstrates that, while there are similarities between the two HMM approaches, there are also distinct characteristics to each of them.

## Discussion

We show that large-scale networks in resting-state MEG can be well described by repeated visits to short-lived transient brain states. Here a state is defined as a distinct spatially and spectrally defined pattern of network activity across the set of considered regions, which span the whole brain. These patterns of activity and phase-coupling were found to be largely symmetric across hemispheres and corresponded to plausible functional systems, including sensory, motor and higher-order cognitive networks. According to our meta-analysis, two higher-order cognitive brain states (or networks) contained regions suggesting a subdivision of the DMN. These subdivisions have characteristic signatures in distinct frequency bands, with one state corresponding to a posterior network with high power and coherence in the alpha range (8–12 Hz) and the other to an anterior network with high power and coherence in the delta/theta range (1–7 Hz). Of note, although we have paid closer examination to the four states represented in Fig. 2, these occupy on average only 30% of the total scanning time (see also Supplementary Fig. 6). While each state has its own subject-specific temporal characteristics (i.e. the state time courses), their spatial and spectral features are defined at the group level. However, note that if they are needed, e.g. to investigate their between-subject spatial and spectral differences, it is straightforward to re-compute subject-specific states features by combining the state time courses and data for each subject separately. Importantly, the proposed HMM is not a biophysical model but a useful perspective on the data. Other models and techniques, such as multivariate autoregressive modelling, offer their own complementary and useful descriptions of the dynamics in the data (see Supplementary Discussion). For further discussion about reproducibility of the results, the estimation of phase, the choice of

the number of states, the state exclusivity assumption and the effect of volume conduction, we refer to the Supplementary Discussion and Supplementary Note 1, 2, 3 and 4.

In previous work on resting-state MEG, an HMM was used to identify fast transient brain states characterised by co-modulations in power[39]. These states corresponded well with canonical resting-state networks in fMRI, showing states switching on ~100–200 ms timescales. However, being based on band-limited power time courses, it was unable to identify potentially faster phenomena that are only apparent in the raw electrophysiological time courses. By contrast, the approach presented in this paper can find brain states with distinct, brain-wide networks of spectrally resolved power and phase-locking from raw MEG time courses. As a consequence, states were found to switch on ~50–100 ms timescales, revealing fast dynamic power and phase-locking information not apparent from a static perspective. Being able to identify phase-coupling is crucial, as this has been proposed as an important mechanism for regulating the integration and flow of cognitive content[6,7]. The identification of large-scale networks of phase-locking in the present work is consistent with the idea that the brain spontaneously evokes the same network dynamics that we see in task[13].

But how can the fact that state visits are often under 100 ms in duration be compatible with the slow frequencies (e.g. delta/theta bands) that characterise the states? For example, an 8 Hz theta cycle, which is in the realms of the frequencies reported here, has a period of 125 ms. This can be reconciled by noting that, although we do not in general capture prolonged oscillations, spectral estimation does not actually require entire cycles. Unlike sliding window approaches, the HMM provides a large number of separated sub-cycle wave segments, with which the spectral estimation at the slow frequencies is possible. This is because frequency is defined instantaneously and depends on the gradient on the signal (see Supplementary Discussion, and Supplementary References), which is theoretically defined at each time point. We have performed simulations to show this empirically (see Supplementary Note 1).

Our results identified two higher-order cognitive states or networks that showed particularly high power and coherence in comparison with the other states. These higher-order cognitive networks also exhibited different temporal dynamics in their state occurrences, notably with longer periods of times between state visits. One of these states represents a posterior network including PCC, precuneus and bilateral intraparietal regions. The other encompasses anterior areas including mPFC and temporal poles, exhibiting strong connectivity with the PCC. Consistently with previous studies in fMRI[40], these results afford the interpretation of the DMN being separable into anterior and posterior subdivisions. The present work, however, offers an important new insight into the electrophysiological properties of these distinct subnetworks. Here the two subdivisions are distinguished from each other by operating within very different frequency bands: in the alpha band for the posterior network, and in the delta/theta band in the anterior network. Furthermore, the PCC may be acting as a link between the two higher-order systems, given that it is present in both networks and has particularly strong delta/theta band connectivity with the mPFC in the anterior higher-order cognitive state. This is consistent with previous work where, based on band-limited power correlations yet ignoring phase-coupling, the PCC was proposed to serve as a hub[41]. The operation of these large-scale cortical phase-coupling networks in very different frequency bands may reflect the different intrinsic timescales that they specialise in within the temporal domain[42–44].

Despite having been attributed a key role in the resting state, and in particular in the DMN[32], the PCC has been somewhat under-represented in the resting MEG/EEG literature, possibly due to the relatively low signal-to-noise ratio (and, hence, visibility) in MEG/EEG[45]. For example, previous analyses of resting MEG data reported putative DMN networks that did not include the PCC[15,34,39]. One possible reason is that the PCC's role as a hub potentially involves many different network states in such a way that is suppressed when examining differences between networks or states[39]. Notably, in a resting MEG study that used time windows of high band-limited power correlation between nodes of the DMN, the PCC exhibited the highest of these correlations[41]. Here the PCC is highly visible when networks are characterised by phase-coupling, especially in the anterior and posterior higher-order cognitive states. Indeed, given the issues in representing the PCC in MEG, it is often merged with precuneus. However, these are remarkably different regions, with different structural connectivity profiles and distinct functional roles[46,47]. Here we used a parcellation that specifically separated the PCC, the anterior precuneus and the posterior precuneus. Supplementary Fig. 5 shows power and phase-coupling with mPFC for each state and each of the three regions. Some differences can be clearly recognised between regions and, in particular, between the PCC and the two precuneus regions. Remarkably, coherence with mPFC is three times higher for the PCC than the precuneus (both anterior and posterior) in the anterior higher-order cognitive state. Also, in the anterior higher-order cognitive state the PCC exhibits strong activity in the delta/theta frequency mode, whereas the precuneus does not.

With the exception of the higher-order cognitive functions, for which we have conducted a meta-analysis against existing literature, our state labelling is purely based on the anatomical location of power and functional connectivity. For example, we refer to the "visual" state as the state with wideband power and connectivity in occipital areas. However, when looking at the spectral characteristics of this state, we find that this activity is primarily occurring in the alpha band (see Supplementary Fig. 4). Given the hypothesised inhibitory role of alpha[48,49], it is likely that this state is cognitively representing a reduction, rather than

an increase, in visual activity. Some further connections to existing literature are plausible. For example, given its long-distance connections between the anterior temporal lobes and its low-frequency dominance, a relation between the anterior higher-order cognitive state and memory retrieval is very likely[12,50,51], yet cognitive control could also be involved[52]. Likewise, the posterior higher-order cognitive state could be related to attention`and cross-modal processing[53]. A separate question is about the mechanisms and causes of state switching and whether these can be linked to avalanches of activity[54].

Our approach is not the first in proposing a segmentation of electrophysiological time series into a discrete set of states. For example, Rabinovich and colleagues[55], among others, argue for the characterisation of brain dynamics as "a task-dependent sequential activations metastable states, that is, states where system variables reach and temporary hold stationary values"; see also ref.[56] for a general reference about metastability in the brain. A prominent related methodology is the EEG microstates framework; see ref.[57]. Segmentation of EEG scalp maps into microstates is based on finding repeating distributions of power across multiple sensors and therefore could be expected to capture interactions related to those that drive the HMM. Microstates have also offered new insights into the nature of resting-state networks including some evidence of a fragmentation of the DMN into anterior and posterior states[58,59]. However, some fundamental differences exist. Most importantly, the HMM directly identifies states with distinct spectral and cross-spectral profiles, including coherence networks in distinct frequency bands and, potentially, at diverse phases[60]. In contrast, while microstates can capture broadband spectral phenomena, their estimation (performed in sensor space) is not based on spectral profiles (but see ref.[61]), and assumes zero-lag (or 180°) phase differences. Also, whereas the proposed model operates in source space, microstates are estimated in sensor space. Although HMM states can also be estimated in sensor space, source reconstruction is, however, useful for noise removal and to better balance the contribution of deeper regions compared with more dominant (superficial) cortical areas. In summary, it is through the use of the HMM that we have been able to reveal that the fragmentation of the DMN into anterior and posterior states is characterised by not only the presence of phase-locking networks but also spectral power and phase-locking in distinct frequency bands. Furthermore, while other approaches applied to EEG data that do capture spectral differences have also been proposed, these are also in sensor space, and so it is more difficult to capture the changes in phase-coupling between specific subnetworks of cortical regions that we find in this work[62,63].

This study focused on lower frequency bands (1–45 Hz). Because of the methodological considerations discussed in the Methods section, and because of the higher signal-to-noise ratio in lower frequency bands (1–30 Hz), low gamma frequencies (30–45 Hz) did not reveal any clear state-specific differences. However, we would expect there to be different patterns in gamma, given the possible top–down modulation of these frequencies by the slow frequencies[64]. Because of the crucial importance of gamma in cognition, having a key role in information transference between regions and plasticity[5,6,65–67], it is of primary interest to understand how gamma frequency is modulated at the whole-brain level across different states. This will be an important area for future studies.

In summary, we have proposed an analysis approach that allows the investigation of dynamic changes in whole-brain phase-coupling in the resting state. Our study revealed that, at these fast timescales, higher-order regions within the DMN dissociate into two spatially, temporally and spectrally distinct states. These states potentially index different higher-order cognitive

processes that themselves operate at different timescales. Although we have focused on this particular aspect of the data, the wealth of information contained in the model output opens many avenues for future analyses, hypotheses and questions. These include the dynamics of specific phase relations between areas, the whole-brain dynamics of gamma at rest and the existence of changing patterns of communication between processes operating at different frequencies.

## Methods

**Data and preprocessing.** As part of the UK MEG Partnership, 77 healthy participants were recruited at the University of Nottingham. All participants gave written informed consent and ethical approval was granted by the University of Nottingham Medical School Research Ethics Committee. A final cohort of 55 participants (mean age 26.5 years, maximum age 48 years, minimum age 18 years, 35 males) was selected for analysis, discarding 22 subjects because of excessive head motion or artefacts. To avoid effects of tissue magnetisation, MEG data were acquired prior to participants entering the MRI. Resting-state MEG data were acquired using a 275-channel CTF MEG system (MISL, Coquitlam, Canada) operating in third-order synthetic gradiometry configuration, at a sample frequency of 1200 Hz. MRI data, used here for the purpose of MEG coregistration, were acquired using a Phillips Achieva 7 T system. (See ref.[68] for further details about MEG and MRI acquisition). MEG data were then downsampled to 250 Hz using an anti-aliasing filter, filtered out frequencies <1 Hz and source-reconstructed using LCMV beamforming[24] to 42 dipoles covering the entire cortex excluding subcortical areas (MNI coordinates are shown in Supplementary Table 1). Thirty-eight of these dipoles were obtained from a ICA decomposition on resting-state fMRI data from the Human Connectome Project, used previously to estimate large-scale static functional connectivity networks in MEG[18]; the other four parcels correspond to the anterior and posterior precuneus that we wanted to disambiguate from the PCC given the importance of this region in the resting state and the left and right intraparietal sulci. Bad segments were removed manually and correction for spatial leakage was applied using the technique described in ref.[25]. The effect of using alternative methods for leakage reduction is discussed in Supplementary Note 2. In order to project the results to brain space, we used a weighted mask, where each region had its maximum value at the centre of gravity.

**The Hidden Markov Model.** As a general framework, the HMM assumes that a time series can be described using a hidden sequence of a finite number of states, such that, at each time point, only one state is active. In practice, because the HMM is a probabilistic model, the inference process acknowledges uncertainty and assigns a probability of being active to each state at each time point. Effectively, this amounts to having a mixture of models (or states) explaining the data at each time point, where the mixture weights are the state probabilities. Importantly, the probability of a state being active at time point $t$ is modelled to be dependent on which state was active at time point $t-1$ (i.e. it is order-one Markovian). The model then assumes that the data observed in each state are drawn from a probabilistic observation model. The observation distribution is of the same family for all states, whereas the observation model parameters are different for each state. The different varieties of the HMM are thus given by which family of probabilistic observation distribution is chosen to model the states. This is useful because different observation distributions can be adequate for different data modalities[22,36,39,69] while preserving a common framework. This can facilitate integration of results across modalities. The variety of the HMM introduced in this paper is presented in the next section along with some theoretical and practical discussion about its properties. Whichever choice of the HMM state distribution, the model can be applied to each subject independently or to the concatenated data of all subjects, such that a group estimation of the states may be obtained. In this paper, the states were defined at the group level; however, the information of when a state becomes active (i.e. the state time course) is still specific to each subject. Inference on the model (i.e. the estimation of the parameters of the posterior distribution; see Fig. 1b) is carried out using variational Bayes (VB), a method providing an analytical approximation at a reasonable cost by assuming certain factorisations in the posterior distribution; we refer to ref.[69] for further details about the inference scheme. Still, because of the high sampling rate of our MEG data (250 Hz) and the relatively high number of subjects (55), standard VB becomes both time and memory consuming. On these grounds, we used stochastic inference to further alleviate computation time[22] such that an average run would take approximately 5 h using a standard workstation with manageable memory usage. After the inference process, the Viterbi path is computed; this is defined as the most probable sequence of (hard assigned, i.e. non-probabilistic) states and can be analytically computed, given the current estimation of the state observation models, using a modification of the standard HMM state time course inference[21].

The HMM analysis was conducted using the HMM-MAR Matlab toolbox (https://github.com/OHBA-analysis/HMM-MAR), which contains detailed documentation of the tools' usage (https://github.com/OHBA-analysis/HMM-MAR/wiki/User-Guide). Furthermore, a script containing the entire pipeline is also available online (https://github.com/OHBA-analysis/HMM-MAR/blob/master/examples/NatComms2018_fullpipeline.m).

**The TDE Hidden Markov Model.** Here we apply a novel variety of the HMM to raw time courses (instead of power envelopes[39]). This allows us to detect changes not only in power but also in phase-locking. Although this was already the case with the HMM-MAR[69], the multivariate autoregressive (MAR) observation model works optimally with a limited number of regions and does not scale to whole-brain analysis. The reason, beyond computational, is that a MAR model of order $p$ needs $42^2 \times p$ autoregressive coefficients to model data with 42 regions of interest (as we use here). This large number of parameters can result in overfitting. As a consequence, this makes the HMM unable to segment the time series effectively.

In this approach, the TDE-HMM, our definition of observation distribution describes the neural activity over a certain time window using a Gaussian distribution with zero mean (i.e. using the covariance matrix) to model the entire window; this is equivalent to saying that our observation model corresponds to the data autocovariance across regions (sometimes referred to as lagged cross-covariance) within such window. For example, if we use a time window of 60 ms, having time point $t$ assigned to a certain state means that, for our current 250 Hz sampling rate and 42-region parcellation, the activity of the 42 channels over a window of 15 time points centred at $t$ gets described by such state's ($15 \times 42$ by $15 \times 42$) autocovariance matrix. This multivariate autocovariance matrix (as the MAR model) can effectively capture patterns of linear synchronisation in oscillatory activity for those time points when a particular state is active, i.e. our model can describe state-wise phase-locking. This is mathematically equivalent to using a standard HMM with a Gaussian observation model on an "embedding" transformation of the original data (see Supplementary Fig. 1 for an illustration of the entire pipeline). In our case, with 55 subjects and 5 min of data at 250 Hz per subject, this amounts to running the HMM on a large (4,125,000 by 630) volume of data. As a result of the computational advantages of stochastic inference[22], it is still possible to handle such large amounts of data. However, it requires estimating $(630 \times 629)/2 = 198,135$ parameters within the multivariate autocovariance matrix per state, which, above and beyond computational considerations, can also lead to severe overfitting problems. To avoid this issue, we ran the HMM on a principal component analysis (PCA) decomposition of the "embedded" space. This not only greatly reduces the complexity of the state distributions but also naturally focusses the slower frequencies in the data. This is a consequence of PCA aiming to explain the highest possible amount of variance in the time series, in combination with the $1/f$ of electrophysiological data (i.e. that most of the power, or variance, is concentrated in the slow frequencies). In particular, we use twice PCs as the number of channels (i.e. 84 PCs). In this data set, this explains on average 60% of the variance (lowest and highest across subjects are, respectively, 55% and 66%). Note that, given that the time series from the source-space parcellation are orthogonal after leakage correction[25], the PCA step can only leverage autocorrelations and non-zero lag cross-channel correlations to achieve an optimal decomposition. Since the non-zero lag cross-channel correlations are very small in comparison with the within-channel autocorrelation of the data[69], we chose a number of PCA components that is a multiple of the number of channels; otherwise, because of the very nature of PCA, the "extra" PCA components will be explaining variance from just a few channels. Precisely which channels is mostly arbitrary, given that all channels were standardised to have the same variance. For example, for our 42-region parcellation, using 100 PCA components will result in $(100 - 42 \times 2 =)$ 16 PCA components explaining variance from a (mostly) random subset of 16 regions.

Therefore, besides the number of states (see Discussion), the important parameters of the model are the length of the window (i.e. the number of lags to be modelled by the state autocovariance matrices) and the number of PCA components. From a practical perspective, a trade-off between these two parameters will prescribe which frequencies in the data the TDE-HMM will be more sensitive to. That is, longer windows (more extended lags) and fewer PCs will incline the model to be more sensitive to the lower frequencies, whereas if we include more PCs and/or reduce the window we will be better able to capture high-frequency differences. In this paper, as mentioned earlier, we used a window of 60 ms and $42 \times 2$ PCs. This window contains one cycle at exactly 16.6 Hz, but note that this does not preclude the model from accessing slower frequencies (see Discussion).

**Source-reconstructed dipole ambiguity.** It is an acknowledged issue that source-localised EEG and MEG data have an arbitrary sign as a consequence of the ambiguity of the source polarity. As source reconstruction, in this case through beamforming[24], is done for each subject separately, the sign of the reconstructed dipoles risks being inconsistent across subjects. This is not a problem when modelling the power time courses but is a cause of concern for models based on the raw signal because connectivity between any pair of regions can cancel out at the group level if regions have their time courses flipped for a subset of the subjects. Here we extend and generalise the basic idea in ref.[69] to multiple leakage-corrected channels, making the assumption that the lagged partial correlation between each pair of brain regions, across several different lags, has the same sign across subjects. (We choose to use partial correlation instead of simple correlation because this is a direct measure, i.e. there are no other channels interfering in the "sign relation"

between every pair of channels.). More explicitly, for all lags (for example, in between $\alpha = -10$ and $\alpha = +10$), we aim to find a combination of sign flips for each subject such that the function

$$\text{Gain}(f) = \sum_{j1,j2} \sum_{\alpha} \left| \sum_s \rho(f_{s,j1} x_{s,j1,0}, f_{s,j2} x_{s,j2,\alpha})/N \right| \qquad (1)$$

is maximised. Here $s$ cycles through the $N$ subjects, $j1$ and $j2$ cycle through regions, $x_{s,j1,\alpha}$ represents the data time series for subject $s$ and source $j1$ that have been lagged $\alpha$ time points, $f_{s,j1}$ takes the value $-1$ or $+1$ and represents whether channel $j1$ is flipped for subject $s$, $\rho()$ represents the partial correlation between a pair of time series and $|\ |$ is absolute value. The idea is that, provided the aforementioned assumption, Gain($f$) will be maximised when the signs are correctly aligned. For example, if there is a strong genuine anti-phase relationship (leading to negative correlation) between a given pair of regions, the sign for these regions will be pertinently flipped for those subjects having an in-phase relation (leading to positive correlation) such that the negative correlations do not get partially cancelled out by the occasional positive correlations when averaging across subjects.

To find the best combination of sign flips such that Gain($f$) is maximised is an integer programming problem and thus finding an exact solution is NP-hard. The computationally expensive step is to compute the (no. of channels by no. of channels) partial correlation matrix for each subject and lag (in our data, 21 lags × 55 subjects matrix inversions of size 42 regions × 42 regions). Once this is computed, it is relatively inexpensive to evaluate the function Gain($f$) given the equivalence

$$\rho(X_{s,j1,0}, -X_{s,j2,\alpha}) = -\rho(X_{s,j1,0}, X_{s,j2,\alpha}). \qquad (2)$$

Therefore, we can afford to evaluate many different solutions, for example by multiple instantiations of a greedy algorithm with random initialisations of the signs. Although in this paper we limited ourselves to this simple approach, other more sophisticated search procedures can easily operate on this scheme.

**Extracting spectral information**. Once the HMM has found the states on the basis of the data's transient spectral properties, a logical further question is how can we extract and represent the states' spectral properties in an informative way. We can extract the spectral information (power and coherence) from the multivariate autocovariance matrix in each state's observation model as it has a direct correspondence to the parameters of a MAR model, which contains the spectral information in the system. However, this estimation is biased towards the low frequencies due to the PCA dimensionality reduction step discussed in the previous section. So that we can effectively access high frequency information, we instead made use of the state-wise multitaper approach introduced in ref.[69], which will provide us with power and spectral coherence for each frequency bin (chosen to be in between 1 and 45 Hz) and state without any PCA-induced bias.

Once we have estimated power and spectral coherence for each state, we factorise this information into different components or frequency modes for ease of interpretation and visualisation (see below about the choice of spectral coherence to quantify phase-coupling). We could use the traditional frequency bands for this matter but instead we opted for estimating these in a data-driven fashion. To do this, we constructed a matrix by concatenating the spectrally defined coherence (the spectral feature that is more interesting for our purposes) across all states and pairs of regions. We shall denote this matrix as $A$. More specifically, $A$ has (12 states × 861 pairs of regions =) 10,332 rows and 90 columns, 90 being the number of frequency bins that we obtained from the multitaper analysis. We then applied a non-negative matrix factorisation (NNMF) algorithm[70] on $A$, asking for four components, which we found to render stable decompositions while still being reasonably frequency specific. This choice of four components corresponds to the coarseness of classical frequency bands often used in the low frequency range we are studying (i.e. low gamma, beta, alpha, delta/theta). NNMF aims to find a factorisation $A = WH$, where $W$ has dimension (10,332 by 4) and $H$ has dimension (4 by 90), such that all the elements in $W$ and $H$ are positive. Each row of $H$ then represents the spectral profile of this component, inferred from the data. These components turn out to roughly correspond to the canonical delta/theta, alpha, beta and (lower) gamma bands. Of these, our interpretations are focussed on the first three components (displayed in Fig. 4, left panels), excluding the fourth (gamma band) component for being potentially less relevant to understanding large-scale synchronisation. Having four frequency modes allowed us, however, to have beta separated from gamma, providing a cleaner view on the data. With the component spectral profiles $H$ in hand (referred to as frequency modes throughout the paper), it is straightforward to obtain values of coherence for each state, pair of regions and NNMF component. We do so by simply multiplying the respective (1 by 90) vector of coherence values by the corresponding transposed row of $H$ (90 by 1). For power, we follow the same procedure, reusing the component spectral profiles that we computed for coherence. Wideband results (Fig. 2) correspond to a simple average across all frequency bins.

For the purposes of visualisation, in Fig. 2, Fig. 4, Supplementary Fig. 2 and Supplementary Fig. 4 we showed only the functional connections that were the strongest in absolute value. To avoid setting an arbitrary threshold, we separately fitted, for each state and NNMF frequency mode (and wideband), a mixture of two Gaussian distributions to the population of functional connections, such that we only show the connections that belong to the Gaussian distribution representing the strongest connections. When the population of functional connections is well represented by a single Gaussian distribution, that is indicative that there are no connections that are pronouncedly stronger than the average connectivity within the state, in which case we do not show any. For reference, the distributions of connectivity values together with the fitted Gaussians are shown in Supplementary Fig. 7 for the examined states. For the wideband results (Fig. 2, Supplementary Fig. 2 and Supplementary Fig. 8), the power maps were thresholded such that only the 50% of voxels with the highest activation or deactivation are shown. For the frequency-specific results (Fig. 4 and Supplementary Fig. 4), the threshold was set to 10%.

This analysis is designed to find which functional connections stand out from a background level of connectivity within each state. A different question is which functional connections, or power increments, are significantly stronger for any given state with respect to the other states. We performed non-parametric statistical testing to investigate these differences, for which we calculated the spectral information (power and connectivity) for each subject separately and then used this between-subject variability to run standard permutation testing analysis. In detail, we used a shared set of permutations for each state, power value and functional connection. At each permutation, we shuffled the target power or functional connection value across states. By running 5000 permutations, we effectively created (for each power and functional connection value) a null distribution of differences between each state's value and the mean value of the other states, which we then used to produce a $p$ value per activation value and functional connection. Supplementary Fig. 3 shows statistically (uncorrected) significant power and functional connectivity increments given a significance level of 0.01.

**How similar are two states?**. Some of the validations carried out in the paper (e.g. about reproducibility of the results) are dependent on comparing different states. Given that each state is characterised by a (time lags×number of regions by time lags×number of regions) autocovariance matrix, we quantified the dissimilarity between states using the concept of Riemannian distance between their corresponding autocovariance matrices, which can be defined as a mathematically rigorous generalisation of the Euclidean distance for dealing with positive definite matrices. Given two covariance matrices $C_1$ and $C_2$, the Riemannian distance is defined as the square root of the sum of the logarithms of the eigenvalues of the product $C_1 * C_2$:

$$d_{\text{riemann}} = \left( \sum \log \ \text{eig}(C_1 * C_2) \right)^{1/2} \qquad (3)$$

**Data availability**. The complete data set, acquired in Nottingham in the context of the MEG UK Partnership, is not currently available as it contains data from human participants including structural scans. The data are held by the MEG UK Partnership, and access to the MEG UK Database can be requested at http://meguk.ac.uk/contact. Preprocessed (and parcellated) data containing the time series as they were fed to the HMM can be accessed at https://ora.ox.ac.uk/objects/uuid:2770bfd4-6ab8-4f1e-b5e7-06185e8e2ae1.

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

## Acknowledgements

We thank Jeroen Van Schependom for his valuable help. This study was supported by an MRC UK MEG Partnership Grant (MR/K005464/1), a James S. McDonnell Foundation Understanding Human Cognition Collaborative Award (220020448) and the NIHR Oxford Health Biomedical Research Centre. The Wellcome Centre for Integrative Neuroimaging is supported by core funding from the Wellcome Trust (203139/Z/16/Z).

D.V. is supported by a Wellcome Trust Strategic Award (098369/Z/12/ Z). M.W.W. is supported by the Wellcome Trust (106183/Z/14/Z) and the MRC UK MEG Partnership Grant (MR/K005464/1). L.T.H. is supported by a Henry Dale Fellowship (208789/Z/17/ Z) from the Wellcome Trust, a NARSAD Young Investigator Grant from the Brain and Behavior Foundation and by the NIHR Oxford Health Biomedical Research Centre. A.C.N. is supported by a Wellcome Trust Senior Investigator Award (ACN) 104571/Z/ 14/Z. B.A.E.H. is funded by an MRC Partnership Grant MR/K005464/1 and an MRC Doctoral Training Grant MR/K501086/1. Data were collected in Nottingham in the context of the MRC-funded MEG-UK partnership. The views expressed are those of the authors and not necessarily those of the NHS, the NIHR or the Department of Health.

## Author contributions

D.V. and M.W.W. designed the study, developed the method and wrote the manuscript; B.A.E.H. collected the data; L.T.H. and A.C.N. provided assistance with interpretation; all the authors edited the manuscript.

## Additional information

**Competing interests:** The authors declare no competing interests.

