## [Peer Review File · Nature Communications]

Reviewers' comments:

Reviewer #1 (Remarks to the Author):

Review for: "Spontaneous cortical activity transiently organises into frequency specific phase-coupling networks", Nature Communications

This is a focused concise paper which addresses a topic of great current interest, and will appeal to a broad readership. The base methods are sound and well-tested, and are deployed here in a novel, data-driven analysis to generate findings that advance the subfield. The authors argue that their data show that frequency-specific oscillations play a role in coordinating multiple functional subnetworks, producing a sequence of states identified via changes in spatially-distinct patterns of power and phase coupling. Furthermore:

- Paper provides much needed data on the transient nature the spontaneous evolution of a sequence of brain states, with a quantitative characterization of these changes over time and modeling (HHM).
- Focus on including inter-area phase-coupling in HMM is key advance, given the different time scales of phase coupling changes and power changes, and the direct theoretical work linking phase coupling to communication efficacy.
- Adaptive and data-driven rather than beginning with fixed but arbitrary parameters
- Novel analysis of the default mode network which suggests a subdivision of at least two higher-order cognitive brain states/active-networks, identified by robust spatial and spectral differences.
- Shows different timescales for state changes in higher-order cognition networks vs. sensorimotor/visual networks.
- Shows frequency-specific subdivision of at least 2 higher-order cognition networks.
- Elegant analysis pipeline that starts from raw MEG time-courses with minimal parameter selection or fine tuning by researcher.

However:

- Needs some light rewriting to clarify what is empirical data, what is theoretical frame and mathematical model, and what are inferred causal mechanisms.
- Explain relation of this method to existing literature of EEG microstates (eg data-driven HHM identification). http://www.scholarpedia.org/article/EEG_microstates
- Code is the best documentation of methods. I did not see link to code posted online; if complete codebase for this specific analysis is not yet ready, include link to the repository which will eventually have it.
- Remaining question, what is the relative weight of power and phase-coupling in defining or clustering data into k distinct states?

Minor points:

L14 Abstract, "ongoing functional brain networks"

Consider "ongoing activity in functional brain networks", "functional brain networks displaying spontaneous ongoing activity", or a similar recast. The network is not ongoing, the activity is, or the output from an operationally-defined method detecting functional connectivity within the network. Also, more emphasis that the contrast is between goal-directed cognitive tasks and spontaneous ongoing activity.

L14 Abstract, "should also be able to"

Consider "would also exhibit the oscillatory signatures associated with transient coordination", or shift "in a similar manner" to "via similar mechanisms for transient coordination" or similar. I am requesting

a clearer distinction between functional role, detected empirical signatures or signal, and hypothesis concerning causal mechanisms. "able to reorganize" strikes me as goal-directed function, whereas I suspect you are predicting that goal-directed vs. spontaneous activity will exhibit similar empirical signatures (namely, transient frequency-specific oscillatory activity and phase-locking).

L15 Abstract "use" -> "developed" ?

L18 Abstract "spatially distinct power"
Consider "spatially-distinct patterns of oscillatory power"

L25 "networks operate"
This may be too strong a statement. You show that the MEG signals associated with ongoing activity in default mode networks can be well characterized as a robust sequence of brain states. You hypothesize that this sequence of states is caused by mechanisms of transient coordination that enable the network to operate and perform functional roles. Recommend rewriting this to make these distinctions clearer.

L55 "distinct spectral and phase-coupling features"
Consider "distinct spatial and spectral features that include both power and phase-coupling" or similar. Spectral includes frequency-specific power and phase, with phase coupling determined by absolute phase in each area.

L57 "For the first time"
Consider adding to abstract to make clear original claims to skimming reader.

L75 "in a completely data-driven way"
I think this should be emphasized before results section, prior work in this area almost always hand-pick parameter values, this is a real advance methodologically.

L524 Space permitting, suggest additional refs for instantaneous frequency:
Cohen L. (1995). Time-frequency analysis. [BOOK]
Gardner TJ and Magnasco MO. (2006). Sparse time-frequency representations. PNAS, 103, 16. doi: 10.1073/pnas.0601707103

L552 Space permitting, suggest additional refs for single neuron selectivity for different time scales:
Canolty et al. (2010). Oscillatory phase coupling coordinates anatomically-dispersed functional cell assemblies. PNAS, 107, 40, 17356-17361. doi: 10.1073/pnas.1008306107

Space permitting and if theory agrees, suggest ref to work of Beggs JM on neuronal avalanches:
http://www.scholarpedia.org/article/Neuronal_avalanche
This is one possible mechanism to provoke changes in brain states.

The methods look sound and I did not see anything that jumped out of me as red flags. However, I am not a direct user of beamformer methods for spatial localization, and I have not used HMMs myself. That said, I felt the methods were well documented and described. Code should be included for final version, especially for any novel code not available elsewhere.

SECTIONS:
Abstract

Introduction

Results

- The states exhibit specific phase-locking connectivity
- Higher-order cognitive states have distinct spectral characteristics
- Higher-order cognitive states have distinct temporal characteristics

Methods

Data and preprocessing

The Hidden Markov Model

The embedded Hidden Markov Model

Source-reconstructed dipole ambiguity

Extracting spectral information

Discussion

Fast Transient Brain States and Slow Rhythms

Subdivision of the Default Mode Network

Posterior Cingulate Cortex in Resting-state MEG

Relationship between Power and Coherence

Gamma-band

Number of Brain States

State exclusivity

Summary

Reviewer #2 (Remarks to the Author):

This is a very interesting paper that dissects spontaneous resting-state brain activity recorded with MEG into transient and intermittent states. The authors suggest that two of them correspond higher-order cognitive states, putatively the anterior and posterior parts of the default mode network (DMN), and others to visual, somatosensory and other networks. The authors find resting-state brain activity to be decomposable into very brief (< 100 ms) states with distinct anatomical patterns and spectral profiles.

I think it is fair to say that the phenomenology of electrophysiological resting-state brain activity, let alone that pertaining to phase correlations, is really quite poorly understood. This paper makes a significant contribution to that understanding and, in my opinion, would influence the field and be interesting to a broad audience. I also applaud the authors' data-driven approach, which I think is fundamentally important for gaining understanding about the large-scale operating mechanics of cerebral activity (much beyond what the hypothesis-driven approach can yield) and thereby also advancing the nascent field of network neuroscience. However, the downside of the authors' innovative and extremely advanced analysis pipeline is that it is also extremely difficult for common readers to understand in terms of actual procedures and their implications and limitations. This could partly be alleviated by expanding and better illustrating the analysis pipeline in supplementary material.

Overall my core concern is that as it is now, the reader must accept the results at face value and has little means for grasping their (neurobiological) validity and (statistical) reliability/robustness. I have outlined below some suggestions that I hope the authors consider to consolidate the manuscript.

Major

1. “Separable neurophysiological states or a continuum of activity artificially split by HMM?” The key premise of this paper is that ‘semi-discrete’ brain states exist and can be identified with the authors’ approach. Since this specific methodological approach has not been validated earlier, it would be good to test how similar states it would find from data that has identical spatiotemporal autocorrelations and 1/f dynamics but no states or community structures per se. I.e., running the same core analyses on well comparable surrogate data would be essential. Vidaurre et al., 2016, NeuroImage paper presented validation for the alt.-hypothesis case where true states were simulated and successfully detected, but I was not able to find validation for the null-hypothesis case. One simple, perhaps adequate, approach would be use temporally randomly rotate 42 dipole time series (preserving temporal autocorrelations and dynamics but breaking inter-areal relationships) and apply forward and inverse transforms to recreate the MEG-related spatial mixing.

2. Reliability and state boundaries. It would be useful to assess the reliability of these observations explicitly. What would be the split-cohort reliability of these state observations? I could not find any indication of how the power maps were thresholded – what is the confidence level for the colored areas to belonging to the state they are assigned to? Visualization of the functional connections is justified by fitting a mixture of two Gaussian distributions to the connection strength distribution: how good are these fits? My guess would be that the joint distribution is a unimodal heavy-tailed one but I would gladly be wrong here. Why not simply estimate the null hypothesis distribution with surrogate data and illustrate connections based on statistical significance?

3. Functional annotations. While the authors mostly write about “higher-order cognitive brain states”, which may be fair given their neuroanatomical localization, several parallels are drawn with the DMN. The authors could match these data with fMRI RSN maps and test quantitatively whether the observed higher-order cognitive states better match the DMN or other networks (see also minor note below about co-localization with prior power-correlation data).

4. Power bias for phase correlation networks. Estimating phase correlations with coherence is fundamentally problematic because, the resulting network is significantly dependent on power so that high-power states have much more weight than low power states in dictating which phase differences “count” in the estimate. As implied in Fig. 3a, this likely leads to an overestimation of how well the phase correlation networks are co-localized with the power topographies as well as to a possible misestimation of what the phase correlation networks really are. The presence of some ‘counter examples’ does not diminish the presence of such a systematic bias. If the authors want to have phase-coupling networks, as opposed to coherence networks, as the theme of the paper, I would urge to use a proper phase synchrony metric, in the simplest by running the coherence function with amplitudes normalized to unity.

5. Brevity of the states: while the authors’ supplementary material suggests that the states’ spectral profile can be technically evaluated with sub-cycle data segments, it remains unclear what can be the physiological meaning of states like this. Is state brevity a “false” result caused by the HMM assumption of just one state being allowed in the entire system at a time in a condition where a hypothetical “truth” is that the states are longer but overlap each other temporally?

Minor

- How are the power maps constructed? Their visualization appears to be in done at the resolution of source dipoles but the HMM analysis and phase correlation networks were done in the 42 parcel parcellation.

- The paper uses a HMM-based approach that has been validated in previous papers, but the 'observables' to the HMM are different to that used in Baker et al. (2014) and Vidaurre et al. (2016). The authors say that the MVAR coefficients that were used as observables in Vidaurre et al. (2016) work well with a few regions, but not many brain regions. In this paper, the observables are 'time-delayed covariance matrices' which are conceptually similar to MVAR coefficients. Please provide some intuition on why MVAR coefficients do not work well when looking at many regions (but time-delayed covariance matrices do). Also since a novel variant of the HMM-based method is used, simulations quantifying its validity would be welcome in the Supplementary section.
- Using time-delayed covariance matrices as observables: please provide information on the maximum lag used and rationale for it, for example in relation to the cycle widths of the observed oscillations on one hand and axonal conduction velocities on the other.
- The authors have set the number of states in the HMM somewhat arbitrarily to 12. Why analyse 12 and proceed to ignore many of them when the 6 state division seems to convey all essential state information noted also for 12 states? Are the states hierarchically organized as in hierarchically modular networks? The authors note that the number of states essentially influences level of detail or resolution at which brain dynamics are viewed. While I agree with this notion, it would be good to provide statistical evidence that more than one stable state exists and is reliable in the first place and further use a measure of stability to quantify whether the chosen numbers of states yield valid/stable divisions into states (analogously to measuring the stability of graph module allocations).
- How do the power topographies of each state match with power-correlation networks reported in the original HMM paper (Baker et al. (2014)) by the same group? While the Baker et al. (2014) paper identified 8 states which corresponded to the different canonical RSNs to some extent, the power topographies (or the phase correlation networks) in this paper do not seem to systematically correspond to the RSNs per se. I do not see that they should, but it would be useful to the community to know whether they do or not. Are the differences simply driven by the power-vs.-raw difference (different observation models) for the HMM-based method?
- Comparison with the new Pascual-Marqui method could be presented quantitatively. Are all Pascual-Marqui connections also found by the Colclough method? For example, if Pascual-Marqui connections are considered a truth network, what is the sensitivity and specificity (or accuracy, TPR, FPR...) of the Colclough network?

Reviewer #3 (Remarks to the Author):

The paper of Vidaurre et al. demonstrates that spontaneous brain activity is parcelled into specific brain states in time and that each of this state is characterized by specific frequency and phase-coupling. This demonstration is based on the analysis of hidden Markov models in source-reconstructed MEG data of 55 subjects.

There is increasing interest in the temporal dynamics of resting-state brain activity and fMRI is limited when it comes to the time scale that is relevant for understanding mental processes. The method presented here is interesting as it allows to resolve this dynamics in time, frequency and phase.

The principle results of the study are:

- a) The life-time of a state is around 50-100 ms.
- b) Power- and phase/locking in specific frequencies characterize these states.

- c) Different states have different dominant frequencies.
- d) The interval times differ for different states.
- e) Frontal and posterior states differ in the dominant frequency.

While these are all interesting observations, they are not entirely new and they largely confirm what has been already observed using other methods applied to EEG or MEG. Unfortunately, the paper makes only little or no contact with this existing literature. In order to make the paper significantly more powerful, the approach and the results should be better embedded in the context of other existing approaches and should be compared to them.

Approaches that should be discussed and compared with are the following:

- a) The EEG microstate approach: A large number of studies over many years have shown that resting-state EEG can be parcelled into states of stable topographies lasting around 100 ms, that are related to large-scale networks seen with fMRI due to scale-free dynamics [1-3]. Topographic time-frequency decompositions have shown that networks contributing to a specific microstates share the same temporal dynamics defined in time, frequency, and phase [4, 5].
- b) Analysis of synchronization patterns of resting state EEG have shown that EEG dynamics consists of a limited number of stable states consisting of core networks that remain stable for about 100 ms and rapidly switch between each other [6].
- c) The general idea of chunking brain dynamics in metastable states has been proposed repeatedly and demonstrated in different ways (see for example the heteroclinic channels of [7] or the coordination dynamics of [8]). Relations between such principles and observed EEG dynamics have been discussed in [9] and more recently in [10].
- d) Also the link to the literature regarding frontal theta and posterior alpha oscillations is missing [11-13]. The results presented in the manuscript, in respect to the localization, network extent and frequency are plausible considering this 'classic' literature concerning EEG oscillations.

In addition to this general request of better embedding the study in the current knowledge on brain state dynamics, the following methodological questions need to be considered:

1. Only four out of twelve observed states are described and discussed in detail in the manuscript. What was the percentage of explained variance for these four states (relative to the total variance)?
2. Power maps are illustrated relative to their temporal average. The scale would be of interest here, i.e. % change or dB in order to appreciate what 'low' and 'high' power means (in numbers). Furthermore, the thresholding criterion for the power maps is missing.
3. Several other important parameters of the method are not justified and not described in detail. Specifically, why were 42 ROIs and 4 frequency profiles chosen?
4. Line 153 -> It is said that frequency bands are split into the classical bands. However, delta/theta is defined as 0.5-10 Hz and alpha as 5-15 Hz. Besides the frequency overlap these are not the classical bands. I assume this is a typo.
5. Line 155 -> Due to the relatively lower SNR in this band and considering the bias towards lower frequency activity resulting from running HMM on PCA components, it is more likely that gamma modes could not be observed given the used methods. What is crucially different from stating that there are no "state-specific differences" for these modes. This issue is addressed in the discussion; however, the wording should also be changed in the results section.
6. Line 317 -> It seems that the authors down-sampled the data before filtering. If so, they do not mitigate the distortion due to aliasing. I assume it is a typo.
7. Line 328 -> By using the hidden Markov Model authors assumed that at each time point only one state is active. Moreover, the states are defined at group level and then the activation of a state at the subject level (Line 344). How do the authors justify that by using a 12-state model they are able both to explain the subject inter-variability and to guarantee that at each time point only one of this state is active?

8. Line 383 -> Authors applied PCA. It would be important to know the amount of variance explained by the decomposition per subject.
9. Line 479 -> The authors should define "in numbers" what they assume saying "having more probability" .

References

1. van de Ville, D., J. Britz, and C.M. Michel, EEG microstate sequences in healthy humans at rest reveal scale-free dynamics. *Proc Natl Acad Sci U S A*, 2010. 107: p. 18179-18184.
2. Khanna, A., et al., Microstates in resting-state EEG: current status and future directions. *Neurosci Biobehav Rev*, 2015. 49: p. 105-13.
3. Lehmann, D. and C.M. Michel, EEG-defined functional microstates as basic building blocks of mental processes. *Clin Neurophysiol*, 2011. 122(6): p. 1073-4.
4. Koenig, T., F. Marti-Lopez, and P. Valdes-Sosa, Topographic time-frequency decomposition of the EEG. *Neuroimage*, 2001. 14(2): p. 383-90.
5. Studer, D., U. Hoffmann, and T. Koenig, From EEG dependency multichannel matching pursuit to sparse topographic EEG decomposition. *J Neurosci Methods*, 2006. 153(2): p. 261-75.
6. Betzel, R.F., et al., Synchronization dynamics and evidence for a repertoire of network states in resting EEG. *Front Comput Neurosci*, 2012. 6: p. 74.
7. Rabinovich, M.I., A.N. Simmons, and P. Varona, Dynamical bridge between brain and mind. *Trends Cogn Sci*, 2015. 19(8): p. 453-61.
8. Fuchs, A. and V.K. Jirsa, eds. *Coordination: Neural, Behavioral and Social Dynamics*. 2007, Springer: Berlin.
9. Tognoli, E. and J.A. Kelso, The metastable brain. *Neuron*, 2014. 81(1): p. 35-48.
10. de Pasquale, F., et al., Cortical cores in network dynamics. *Neuroimage*, 2017.
11. Jensen, O. and C.D. Tesche, Frontal theta activity in humans increases with memory load in a working memory task. *Eur J Neurosci*, 2002. 15(8): p. 1395-9.
12. Klimesch, W., alpha-band oscillations, attention, and controlled access to stored information. *Trends Cogn Sci*, 2012. 16(12): p. 606-17.
13. Cavanagh, J.F. and M.J. Frank, Frontal theta as a mechanism for cognitive control. *Trends Cogn Sci*, 2014. 18(8): p. 414-21.

We thank the three Reviewers for their critical revisions, which have greatly helped improve the paper. Changes are highlighted in red.

Reviewer #1

Comment 1. *Needs some light rewriting to clarify what is empirical data, what is theoretical frame and mathematical model, and what are inferred causal mechanisms.*

We thank the Reviewer for the suggestion. We have clarified these points further in the last paragraph of the Introduction.

With regard to the empirical data:

"We used resting-state MEG data from 55 healthy subjects, source-reconstructed to 42 regions across the entire cortex. Notably, applying this approach to these data revealed..."

In relation to the theoretical framework:

"To identify the possible presence of these events, we use a new analysis approach based on the Hidden Markov Model (HMM; Rabiner, 1989), a general mathematical framework previously shown to find recurring states in brain data (Vidaurre et al, 2017)."

We note that, strictly speaking, we are not aiming to pinpoint the *causal* mechanisms underlying the states visits and switches. In this paper, we are focusing on describing these states and their main characteristics, relating them to previous literature but without hypothesising explicit causality. This point, while important, requires further extensive research. We have included a new section in the Discussion, "**Biological underpinnings and functional labelling**", where we now further clarify this point.

Comment 2. *Explain relation of this method to existing literature of EEG microstates (eg data-driven HMM identification). http://www.scholarpedia.org/article/EEG_microstates.*

Following the Reviewer's suggestion, we have added the section "Relation to EEG Microstates" in the Discussion, which contains a brief conceptual comparison to the EEG microstates framework:

"Our approach is not the first in proposing a segmentation of electrophysiological time series into a discrete set of states. For example, Rabinovich et al. (2015), among others, argue for the characterisation of brain dynamics as "a task-dependent sequential activations metastable states, that is, states where system variables reach and temporary hold stationary values" (see also Tognoli and Kelso (2014) for a general reference about metastability in the brain). A prominent related methodology is the EEG microstates framework (see e.g. van de Ville et al. (2010); Khanna et al. (2010)). One essential difference between the approach taken in this work and the EEG microstates is that we characterise HMM states in source-space. Most importantly, our states are specifically defined as periods in time where the data exhibits distinct spectral and cross-spectral properties. This allows us to identify states that correspond to networks of specific multivariate spectral patterns, including coherence. By contrast, EEG microstates do not appear to exhibit distinct spectral properties. While other approaches applied to EEG data that do capture spectral differences have also been proposed, these are in sensor space, and so cannot capture the changes in phase-coupling between specific subnetworks of cortical regions that we find in this work (Koenig et al., 2001; Studer et al., 2006; Betzel et al., 2012)."

Comment 3. *Code is the best documentation of methods. I did not see link to code posted online; if complete codebase for this specific analysis is not yet ready, include link to the repository which will eventually have it.*

We fully agree with the Reviewer's comment. Although the code of the toolbox was already online, we have now uploaded the script with the entire analysis pipeline (from preprocessing to the production of the figures) to our Github repository. In the header of the Methods section, we have now included the following text:

"The HMM analysis was conducted using the HMM-MAR Matlab toolbox¹, which contains detailed documentation of the tools' usage². Furthermore, a script containing the entire pipeline is also available online³."

Comment 4. *What is the relative weight of power and phase-coupling in defining or clustering data into k distinct states?*

The Reviewer raises an important question, related to the one addressed in the Discussion "Relationship between Power and Coherence". In order to bring some clarification, we have expanded the Results by adding the Section "Power vs connectivity in driving state switching", which reads now as follows:

"An important question is which features in the data (i.e. power or connectivity) are driving the HMM state segmentation. Given that power can be estimated more precisely than cross-spectral properties (such as coherence), and that the leakage correction procedure (Colclough et al., 2015) may remove some genuine connectivity information (if interactions occur with, or close to, zero-lag), changes in power are expected to drive a considerable amount of state switching. By manipulating the diagonal and off-diagonal elements of the autocovariance matrices that characterise each state, we computed the Riemannian distance (see Methods) between each pair of states using (i) power and coherence, (ii) just coherence, and (iii) just power. **Fig. 7a** reflects that the contribution of power and coherence to state differences is variable, but, on average, the contribution of power is indeed around four times higher on average (note the difference in the scale bar).

To compare our results to previous work, we also ran the HMM on power envelopes computed from the data band-passed filtered between 1Hz and 40Hz (Baker et al., 2014), which defines HMM states as having distinct patterns of power and power correlations in a single frequency band. We then computed the spatial correlation between the power maps and the phase-coupling connectivity profiles (in either case using the multitaper after the HMM inference) between the two types of HMM. We paired the states between the two runs such that the correlations are maximal. **Fig. 7b** shows that some power maps are relatively well correlated between the two runs, and that the differences of functional connectivity between the two runs are in general larger than the differences in power. In summary, this analysis demonstrates that, while there are similarities between the two HMM approaches, there are also distinct characteristics to each of them."

Minor Comment 1. *Abstract, "ongoing functional brain networks"*

Consider "ongoing activity in functional brain networks", "functional brain networks displaying spontaneous ongoing activity", or a similar recast. The network is not ongoing, the

¹ <https://github.com/OHBA-analysis/HMM-MAR>

² <https://github.com/OHBA-analysis/HMM-MAR/wiki/User-Guide>

³ https://github.com/OHBA-analysis/HMM-MAR/blob/master/examples/NatComms2018_fullpipeline.m

activity is, or the output from an operationally-defined method detecting functional connectivity within the network. Also, more emphasis that the contrast is between goal-directed cognitive tasks and spontaneous ongoing activity.

We thank the Reviewer for this and the following comments, which we have incorporated to the text.

We have changed this now to "the ongoing activity ascribed to the different functional brain networks"

Minor Comment 2. *Abstract, "should also be able to"*

Consider "would also exhibit the oscillatory signatures associated with transient coordination", or shift "in a similar manner" to "via similar mechanisms for transient coordination" or similar. I am requesting a clearer distinction between functional role, detected empirical signatures or signal, and hypothesis concerning causal mechanisms. "able to reorganize" strikes me as goal-directed function, whereas I suspect you are predicting that goal-directed vs. spontaneous activity will exhibit similar empirical signatures (namely, transient frequency-specific oscillatory activity and phase-locking).

We have changed this to "should also be able to reorganise and coordinate via similar mechanisms"

Minor Comment 3. *Abstract "use" -> "developed"?*

We have now included this change.

Minor Comment 4. *Abstract "spatially distinct power". Consider "spatially-distinct patterns of oscillatory power"*

We have now included this change.

Minor Comment 5. *"networks operate". This may be too strong a statement. You show that the MEG signals associated with ongoing activity in default mode networks can be well characterized as a robust sequence of brain states. You hypothesize that this sequence of states is caused by mechanisms of transient coordination that enable the network to operate and perform functional roles. Recommend rewriting this to make these distinctions clearer.*

We have changed it to "have characteristic signatures"

Minor Comment 6. *"distinct spectral and phase-coupling features"*

Consider "distinct spatial and spectral features that include both power and phase-coupling" or similar. Spectral includes frequency-specific power and phase, with phase coupling determined by absolute phase in each area.

We have now included this change.

Minor Comment 7. *"For the first time"*

Consider adding to abstract to make clear original claims to skimming reader.

Because we already used this expression in the Introduction ("For the first time, this allows for the identification of brain-wide networks"), we preferred to omit it in the Abstract. We

however thank the Reviewer for the suggestion.

Minor Comment 8. *"in a completely data-driven way"*

I think this should be emphasized before results section, prior work in this area almost always hand-pick parameter values, this is a real advance methodologically.

We have added it to the abstract: "To test this hypothesis, we **developed** a novel method for identifying, **in a completely data-driven way**, repeating patterns of large-scale phase-coupling network dynamics"

Minor Comment 9. *Space permitting, suggest additional refs for instantaneous frequency:*

Cohen L. (1995). Time-frequency analysis. [BOOK]

Gardner TJ and Magnasco MO. (2006). Sparse time-frequency representations. PNAS, 103, 16. doi: 10.1073/pnas.0601707103

We have added the second reference. Unfortunately we are over the limit in number of references allowed by the journal, so we had to leave out the first.

Minor Comment 10. *Space permitting, suggest additional refs for single neuron selectivity for different time scales:*

Canolty et al. (2010). Oscillatory phase coupling coordinates anatomically-dispersed functional cell assemblies. PNAS, 107, 40, 17356-17361. doi: 10.1073/pnas.1008306107

We have added the reference, and cited it in the first paragraph of the Introduction.

Minor Comment 11. *Space permitting and if theory agrees, suggest ref to work of Beggs JM*

on neuronal avalanches: http://www.scholarpedia.org/article/Neuronal_avalanche

This is one possible mechanism to provoke changes in brain states.

We have added the following to the Discussion:

"... A separate question is about the mechanisms and causes of switching, and whether these can be linked to avalanches of activity (Beggs and Plenz, 2003)."

Reviewer #2

Comment 1. *"Separable neurophysiological states or a continuum of activity artificially split by HMM?" The key premise of this paper is that 'semi-discrete' brain states exist and can be identified with the authors' approach. Since this specific methodological approach has not been validated earlier, it would be good to test how similar states it would find from data that has identical spatiotemporal autocorrelations and 1/f dynamics but no states or community structures per se. I.e., running the same core analyses on well comparable surrogate data would be essential. Vidaurre et al., 2016, NeuroImage paper presented validation for the alt.-hypothesis case where true states were simulated and successfully detected, but I was not able to find validation for the null-hypothesis case. One simple, perhaps adequate, approach would be use temporally randomly rotate 42 dipole time series (preserving temporal autocorrelations and dynamics but breaking inter-areal relationships) and apply forward and inverse transforms to recreate the MEG-related spatial mixing.*

The Reviewer raises the question of to which extent semi-discrete brain states emerge in surrogate data. Following the Reviewer's suggestion we have now added an appropriate analysis. In particular, we simulate from one single observation model (see new section in the Supplemental Information "Simulating data from the HMM" and the response to Minor Comment #2 below), set the HMM to use four states, and run the HMM inference. We now indicate at the end of the "Simulating data from the HMM" section:

"... Furthermore, when simulating from one single state, the HMM inference was able to reduce the complexity of the model by eliminating all states but one."

We have also now added an investigation of how the HMM states might differ between those we find on data simulated from complex single-state dynamic models, and those that we find in the real data. To this end, we have performed a further set of synthetic simulations, and added a new section in the Supplemental Information, with the title "HMM states from surrogate data" and the following content:

"In this paper, we have proposed a model that finds separable, spectrally-defined states from MEG data. Applied on the resting-state, we found a number of states with interpretable characteristics. Being aware that the brain complex dynamics can be equally well represented in multiple ways (see *Alternative representations of the data* above), it is important to investigate how the HMM states might differ between those we find in surrogate data simulated from complex single-state dynamic models, and those that we find in the real data. To test this, we implemented a surrogate data generation procedure where we kept analogous 1/f dynamics and autocorrelations while breaking the state-specific dynamics by using autoregressive models. In particular, using the 42-channels data used in the rest of the paper before leakage correction, we estimated two models: (i) a multivariate autoregressive model of order 3 (MAR(3), with $42^2 \times 3$ parameters) which captured between channel autocorrelations, and (ii) a collection of univariate autoregressive models (one per channel) of order 21 (AR(21), with 21×42 parameters) which did not capture between-channels spectral characteristics, but which was able to estimate more detailed within-channel spectral features. We then sampled data from these two models, corrected for signal leakage, and applied the HMM.

Consistent with our expectations, and as discussed in previous work (Vidaurre et al, 2017b), the autoregressive model is indeed able to represent complex dynamics. Depending on the complexity of the autoregressive model (in particular, in the case of the MAR(3) and AR(21) models), a single (low-rank) lagged cross-correlation (as corresponds to the HMM states in our model) cannot represent the data well enough, and, therefore, several states are necessary. In other words, complex 1/f and cross-spectral dynamics can either be represented by an autoregressive model with a large number of parameters, or a set of less complex models as we use in this work. As a consequence of this, different HMM states emerge from these simulated data."

We note here, that it is important to be clear that the HMM is not a biophysical model, and so alternative analyses of the data can also represent the brain activity well, and from a different perspective. To make this clearer, we also now include a new section named "Alternative representations of the data" in the Supplementary Discussion, reading as follows:

"The TDE-HMM is a useful representation of the data, but is not the only possible one. For instance, a high order multivariate autoregressive model has the potential to explain very rich dynamics to similar extent, but in an alternative manner, to an HMM with a simpler observation model (Vidaurre et al., 2016; Vidaurre et al, 2017b). Armed with just resting data, it is not possible to disambiguate between these two different descriptions of the data. Which one is more appropriate rather depends on the question in hand. A potential reason to use the HMM over a single-state more

complex model (such as a high order multivariate autoregressive model) is that it explicitly parameterises the time series through the state time courses, opening avenues to investigate, for example, the interactions between rest and task. Further, it is through the use of the HMM in this work that we have been able to successfully identify networks of spatially distinct patterns of oscillatory power and phase-coupling in specific frequency bands, in a manner that has not been achieved previously with other approaches, including the autoregressive model.”

Nonetheless, given that the data simulated from the autoregressive model is able to represent complex dynamics and therefore produces states when using the HMM, the question follows as to what these surrogate states represent, and whether the states obtained from real data have different characteristics with the states obtained from the simulated data. We continue in section "HMM states from surrogate data" as:

“We compared the HMM decomposition obtained from the real data and the HMM decomposition obtained from these synthetic scenarios. **Fig. SI-8** shows the results. The states obtained from real data hold significant differences with the states obtained from the MAR(3) and the AR(21) models. As a first approach, we compared the wideband spectral maps (see Methods) between the surrogated and the real states. For this, we paired up the synthetic states with the real states such that the correlation is maximal. In the MAR(3) case, the states have relatively low correlations between real and synthetic (top left); in the AR(21) case, which captures the within-channel spectral information more faithfully, the correlation is much higher, with the interesting exception of the higher-cognitive states and the fifth state (top right). As demonstrated in the bottom-right panel, these high correlations are however trivially explained by the across-states average power profile being very similar between synthetic and real states; this was expected given the AR(21) model’s high explanatory power in the spectral domain. Note that this grand-average correlation is missed in the MAR(3) model case, most likely due to leakage correction removing a large extent of the (stationary) information. Most importantly, the states obtained from the real data are much more diverse than those corresponding to either of the synthetic models, as shown in the bottom panels. This, together with the inability to identify the higher-cognitive states, suggests that the HMM states obtained here are not trivially obtained from just *any data* with the same spatiotemporal autocorrelations.”

Comment 2. *Reliability and state boundaries. It would be useful to assess the reliability of these observations explicitly.*

What would be the split-cohort reliability of these state observations?

We have run half-split replication analyses, where we compared the states obtained separately from two separate partitions of the data. This is summarized in the Discussion, within the rebranded section "Number of states and state reliability":

"Another important aspect (and a sensible way of guiding the choice of the number of states) is state reliability, that is, how robust are the states across, for example, half-splits of the data? We randomly divided the subjects into two partitions and ran the HMM on each of half. We then compared the states between the partitions and to the full data set run. Details of the experiments and results are provided in the **Supplemental Discussion** and **Fig. SI-10**. In short, most states were reproducible across data halves to a large extent, which was quantitatively verified through non-parametric statistical testing. This analysis demonstrated that both the method and the results are robust in this data set; further replication in other data sets and experimental paradigms is however fundamental and will be addressed in the future."

We go into further detail in the Supplemental Discussion (section Reproducibility):

"We assessed the reproducibility of the states by randomly splitting the data into two groups of subjects (half-splits) and running five times the HMM inference separately on each half. We also ran the HMM on the full data set five times. This is intended to evaluate the reproducibility of the results both across different HMM runs and across different subjects. We obtained 12 states from each run and matched the states across runs (between the two partitions and to the full cohort run) such that the similarity between state pairs is maximal. We used Riemannian distances to quantify the dissimilarities between states (see Methods). **Fig. SI-10a** shows the Riemannian distances between each pair of states (within and between runs). We then performed statistical testing on the consistency between runs across halves. Conceptually, we aimed to test if each pair of matched states (one per half) reliably represents the same process. If the distance between the states within the pair is significantly lower than between any two non-matched states, the state represented by this pair is robust across runs and subjects. This is shown in **Fig. SI-10b** (left), where several states appear to be significantly reliable. A notable exception is the anterior higher-order cognitive state, the reason being its high similarity with state 5 (see **Fig. SI-10a**); that is, these two states are relatively similar and can potentially be mixed in certain runs. If, on the other hand, we test the distance of each pair of matched states against the *average* distance between any pair of states (that is, a less conservative test), the anterior higher-order cognitive state appears as highly reliable (**Fig. SI-10b**, right). In summary:

- Overall, there is a strong similarity across runs, both within and between half-split partitions of the data (and to the full data set).
- Within sessions, some states are relatively similar, suggesting some form of state hierarchy."

In terms of the state boundaries and the related assumption of mutual exclusivity, we have added the following to the discussion (section "State occurrence exclusivity"):

"The model specification of the HMM, through assigning state probabilities at each time point, implicitly assumes that only one state is active at each point in time. However, it is worth noting that it is still possible for network multiplexing to be realised at slower time scales through temporal correlation of the rate of occurrence of states. At the faster time-scale of HMM switching, it is important to note that any conclusion about brain network exclusivity must be made with caution, and is by no means necessarily a physiologically meaningful feature of the brain. Addressing the information contained in the state time courses at multiple time scales is an important area for future investigations."

I could not find any indication of how the power maps were thresholded – what is the confidence level for the colored areas to belonging to the state they are assigned to?

Whereas we used a data-driven method (i.e. mixture modelling) to define which functional connections were to be shown in the "Phase-coupling" plots, with the power maps we just selected a fixed percentile of the voxels. In the previous version, this threshold was not consistent between figures. In this version of the manuscript, we have made this comparable across figures. For the wideband maps we show the show the 50% of voxels with the highest departure from the state average (either as an activation or a deactivation). For the frequency-specific results (e.g. Fig. 4) we show the 10%, with the intention of narrowing down into the main features of the maps. The reason of not using mixture modelling for the power maps is because, while there are 861 possible functional connections, the power maps have only 42 elements, which makes Gaussian mixture

modelling less practical. We have clarified the way the maps were constructed in the section "The states exhibit specific phase-locking connectivity":

"The power maps are in relation to the mean power across states."

And in the Methods' section "Extracting spectral information":

"For the wideband results (Fig. 2, Fig. SI-2 and Fig. SI-9), the power maps were thresholded such that only the 50% of voxels with the highest activation or deactivation are shown. For the frequency-specific results (Fig. 4 and Fig. SI-4), the threshold was set to 10%."

Furthermore, we did not perform any statistical analysis in this case, so these colours reflect just the power average. We have however added maps of statistically significant power and connectivity maps in a separate plot - see below.

Visualization of the functional connections is justified by fitting a mixture of two Gaussian distributions to the connection strength distribution: how good are these fits? My guess would be that the joint distribution is a unimodal heavy-tailed one but I would gladly be wrong here.

The referee is right in noting that this distribution is sometimes, yet not always, unimodal with a heavy tail on the right. The intention of using this approach was to avoid arbitrary thresholding, i.e. we did not intend to claim bimodality as an interesting finding. Generally, the Gaussian mixture model fits were found to be qualitatively good. For reference, the distributions of connection strengths are shown in Fig SI-7 for some states.

Why not simply estimate the null hypothesis distribution with surrogate data and illustrate connections based on statistical significance?

This is an important point. Here, the Gaussian mixture model (GMM) analysis is intentionally used to find which functional connections stand out from a background level of connectivity *within each state*, rather than to ask which connections reject the null. To make this clear we have now say in Methods:

"This analysis is designed to find which functional connections stand out from a background level of connectivity *within each state*"

An alternative analysis is the assessment of which connections are significantly stronger for any given state *with respect to the other states*. In order to address the latter question, we have used the between-subject variability in the spectral estimations to perform permutation testing analysis. We have added the results of this analysis in Fig SI-3, where we show the power increments and functional connections that are significantly stronger for these states than for the other states; we now describe this analysis in the Methods section:

"A different question is which functional connections, or power increments, are significantly stronger for any given state *with respect to the other states*. We performed non-parametric statistical testing to investigate these differences, for which we calculated the spectral information (power and connectivity) for each subject separately, and then used this between-subject variability to run standard permutation testing analysis. In detail, we used a shared set of permutations for each state, power value and functional connection. At each permutation, we shuffled the target power or functional connection value across states. By running 5000 permutations, we effectively

created (for each power and functional connection value) a null distribution of differences between each state's value and the mean value of the other states, which we then used to produce a p-value per activation value and functional connection. **Fig. SI-3** shows statistically (uncorrected) significant power and functional connectivity increments given a significance level of 0.01."

Comment 3. Functional annotations. *While the authors mostly write about "higher-order cognitive brain states", which may be fair given their neuroanatomical localization, several parallels are drawn with the DMN. The authors could match these data with fMRI RSN maps and test quantitatively whether the observed higher-order cognitive states better match the DMN or other networks (see also minor note below about co-localization with prior power-correlation data).*

We think this is a really useful suggestion, so we have implemented it and included it in the main Results (new section "The higher-cognitive states and the canonical fMRI DMN"), reading as follows:

"Throughout the paper, we have drawn a link between the two identified higher-cognitive states and the DMN. We now evaluate quantitatively this relation using a meta-analysis technique. In particular, using the system described by Tarkoni et al. (2011), we extracted a canonical map of DMN activation from the fMRI research literature (**Fig. 6a**). Using the wideband power maps (as shown in **Fig. 2**), we computed masks by selecting the 10% most active voxels for each state. We then defined the union of the two higher-cognitive states (depicted in **Fig. 6b**) by selecting the voxels that are active in either of the two corresponding masks. Using these masks, **Fig. 6c** shows the correlation of each state to the canonical DMN map. As observed, the two higher-order cognitive states hold the highest correlation of all states with the canonical DMN, with the union of the two higher-cognitive states being more correlated to the canonical DMN map than any individual state. This result quantitatively demonstrates the relation of the two higher-cognitive states to what is understood by the canonical DMN as measured by fMRI."

We should nevertheless clarify that our annotations were primarily based on anatomical locations and not necessarily cognitive function. For example, the visual state was named after having higher power and connectivity in visual occipital areas. However, this activation is primarily in the alpha range which is rather indicative of inhibition (Klimesch, 2012); therefore, this state might well represent a reduction in visual processing. We have created a new section in the Discussion (**Biological underpinnings and functional labelling**) in order to clarify this point, with the following content:

"In this paper, we have characterised large-scale brain states from MEG data, in terms of power and phase-coupling. In order to gain insight into the mechanistic underpinnings of these patterns and their dynamics, the descriptive methods presented here will need to be combined with biophysical modelling of large-scale networks (Woolrich and Stephan, 2013). With the exception of the higher-order cognitive functions, for which we have conducted a meta-analysis against existing literature, our state labelling is purely based on the anatomical location of power and functional connectivity. For example, we refer to the "visual" state as the state with wideband power and connectivity in occipital areas. However, when looking at the spectral characteristics of this state, we find that this activity is primarily occurring in the alpha band (see **Fig. SI-4**). Given the hypothesised inhibitory role of alpha (Jensen and Mazaheri, 2010; Klimesch, 2012), it is likely that this state is cognitively representing a reduction, rather than an increase, in visual activity. Some further connections to existing literature are plausible. For example, given its long-distance connections between the anterior temporal lobes and its low-frequency dominance, a

relation between the anterior higher-order cognitive state and memory retrieval is very likely (Maguire and Mummery, 1999; Jensen and Tesche, 2002; Fell and Axmacher, 2011), yet cognitive control could also be involved (Cavanagh and Frank, 2014). Likewise, the posterior higher-order cognitive state could be related to attention and cross-modal processing (Behrmann et al., 2004). A separate question is about the mechanisms and causes of state switching, and whether these can be linked to avalanches of activity (Beggs and Plenz, 2003)."

Comment 4. *Power bias for phase correlation networks. Estimating phase correlations with coherence is fundamentally problematic because, the resulting network is significantly dependent on power so that high-power states have much more weight than low power states in dictating which phase differences "count" in the estimate. As implied in Fig. 3a, this likely leads to an overestimation of how well the phase correlation networks are co-localized with the power topographies as well as to a possible misestimation of what the phase correlation networks really are. The presence of some 'counter examples' does not diminish the presence of such a systematic bias. If the authors want to have phase-coupling networks, as opposed to coherence networks, as the theme of the paper, I would urge to use a proper phase synchrony metric, in the simplest by running the coherence function with amplitudes normalized to unity.*

We would like to clarify that the computation of spectral coherence is, within each state, already normalised by the squared magnitude of the spectral power of each signal by definition:

$$\text{Coh}_{ij}(f) = \text{Pow}_{ij}(f) / (\text{Pow}_i(f) \text{Pow}_j(f))^{1/2}$$

Nevertheless, we agree that estimations of time-varying phase-coupling are not rigorously proved to be independent from time-varying power. We can however claim that, by accounting for time-varying power (as the HMM does), we are much better able to reveal the presence of phase-coupled networks.

Note that the abstract is now clearer and consistent with this message:

"To test this hypothesis, we developed a novel method for identifying, in a completely data-driven way, large-scale phase-coupled network dynamics, and show that resting networks in magnetoencephalography are well characterised by visits to short-lived transient brain states, with spatially distinct patterns of oscillatory power and coherence in specific frequency bands."

To make this clearer, we have also added the following to the new section of the Supplemental Discussion (Relationship between power and coherence estimations) dedicated to this question:

"The inference of networks of coherence using the HMM reveals the presence of strong phase-coupling in resting state MEG data. However, while there are examples of coherence and power not changing together, it remains possible that many of the changes in coherence between states could be driven by changes in power. Nonetheless, the evidence presented in this paper suggests that the fast dynamics characterised by the HMM reveal the presence of phase-coupled networks in resting MEG data better than equivalent techniques averaged over slower time-scales.

A relevant related issue is the choice of metric used to measure phase-coupling. There are different alternatives, of which spectral coherence and phase-locking value (PLV) are two popular examples. Although PLV has been claimed to represent phase-coupling more faithfully than spectral coherence (Lowet et al., 2016), in this work we found PLV to be indeed less robust at the large scale than coherence, partly because the application of our data-driven spectral decomposition (as we performed for coherence) is not directly applicable, so we had to rely on an arbitrary specification of the frequency bands. Furthermore, PLV has its own limitations, such as its dependence on filtering and the subsequent use of the Hilbert transform for instantaneous phase calculation; see (Huang et al., 1998) for a comprehensive description of this issue. It is also worth noting that one of the main limitations of spectral coherence is stationarity, which is mitigated here by the fact that the HMM breaks, to some extent, the non-stationarity of the signal into short visits to quasi-stationary states. Further, PLV is also not automatically immune to power bias, given that phase-locking is inferred more reliably when the power (and therefore the signal-to noise ratio) is high. The advantages and disadvantages of different approaches to compute phase-coupling are however beyond the scope of this paper, as a satisfactory solution would require to properly deal with the problems of nonlinearity and non-stationarity, at the heart of the limitations of both spectral coherence and phase-locking value.

Comment 5. *Brevity of the states: while the authors' supplementary material suggests that the states' spectral profile can be technically evaluated with sub-cycle data segments, it remains unclear what can be the physiological meaning of states like this. Is state brevity a "false" result caused by the HMM assumption of just one state being allowed in the entire system at a time in a condition where a hypothetical "truth" is that the states are longer but overlap each other temporally?*

The reviewer raises a good point, and which relates to our response to this Reviewer's Comments 1 and 2. That is, we are not claiming that such short semi-discrete brain states necessarily exist as a unique biophysiological ground truth, or even that the HMM explains all aspects of the data. However, the HMM does offer a useful perspective on the data, allowing us to find meaningful patterns of power and connectivity and state time courses that can be easily interpreted. Please see the response to Comments 1 and 2 for how this is now made clearer in the paper.

More specifically, the values for the state dwell times could indeed be misleading if the exclusivity assumption is strongly violated. To clarify this issue, we have added the following sentence in the Results section 'Higher-order cognitive states have distinct temporal characteristics':

"Note that we have examined only relative differences between the states; since the absolute value of these temporal features must be interpreted with caution given the state exclusivity assumption of the HMM (see Discussion for details)."

Also, as already indicated in the response to Comment 2, we now have the following text in the Discussion:

"The model specification of the HMM, through assigning state probabilities at each time point, implicitly assumes that only one state is active at each point in time. However, it is worth noting that network multiplexing might still be realised at slower time scales through temporal correlation of the rate of occurrence of states. At the time-scale of HMM switching, it is important to note that any conclusion about brain network exclusivity must be made with caution and is by no means necessarily a physiologically meaningful feature of

the brain. Addressing the information contained in the state time courses at multiple time scales is an important area for future investigations."

Minor Comment 1. *How are the power maps constructed? Their visualization appears to be in done at the resolution of source dipoles but the HMM analysis and phase correlation networks were done in the 42 parcel parcellation.*

We note in the Methods section:

"In order to project the results to brain space, we used a weighted mask, where each region has its maximum value at the center of gravity."

Following Reviewer #1's comment, we have included a script with the entire pipeline in our toolbox's Github site [1], including the generation of the power and connectivity maps. We now refer to this from the Methods section:

"The HMM analysis was conducted using the HMM-MAR Matlab toolbox⁴, which contains detailed documentation of the tools' usage⁵. Furthermore, a script containing the entire pipeline is also available online⁶. "

Minor Comment 2. *The paper uses a HMM-based approach that has been validated in previous papers, but the 'observables' to the HMM are different to that used in Baker et al. (2014) and Vidaurre et al. (2016). The authors say that the MVAR coefficients that were used as observables in Vidaurre et al. (2016) work well with a few regions, but not many brain regions. In this paper, the observables are 'time-delayed covariance matrices' which are conceptually similar to MVAR coefficients. Please provide some intuition on why MVAR coefficients do not work well when looking at many regions (but time-delayed covariance matrices do).*

In order to clarify this question, we have now added the following sentences in the 'The Embedded Hidden Markov Model' Section:

"The reason, beyond computational, is that a MAR model of order p needs $42^2 \times p$ autoregressive coefficients to model data with 42 regions of interest (as we use here). This large number of parameters can result in overfitting. As a consequence, this makes the HMM unable to segment the time series effectively."

Also since a novel variant of the HMM-based method is used, simulations quantifying its validity would be welcome in the Supplementary section.

As requested by the Reviewer, we have added a new section to the Supplementary Information, "Simulating data from the HMM", with the following content:

"Each state model of the TDE-HMM corresponds to a Gaussian process (Rasmussen and Williams, 2006). In order to generate T time points of data from a given state, one can construct a (no. of regions \times T) by (no. of regions \times T) covariance matrix by rearranging the elements of the (no. of regions \times time lags) by (no. of regions \times time lags) state covariance matrix. In order to generate data

⁴ <https://github.com/OHBA-analysis/HMM-MAR>

⁵ <https://github.com/OHBA-analysis/HMM-MAR/wiki/User-Guide>

⁶ https://github.com/OHBA-analysis/HMM-MAR/blob/master/examples/NatComms2018_fullpipeline.m

from the TDE-HMM, we used the two higher-order states and selected a subset of regions for computational simplicity (ACC, PCC, and left and right intraparietal sulci). We then sampled 30min of data, alternating between these two states, with state visits set to last from 0.2 to 2s. We then ran the HMM inference on the simulated data. The inference, as shown in **Fig. SI-13**, was able to accurately recover the simulated state time course with a correlation between simulated and estimated time courses of $r=0.98$. Furthermore, when simulating from one single state, the HMM inference was able to reduce the complexity of the model by eliminating all states but one (not shown).

Minor Comment 3. *Using time-delayed covariance matrices as observables: please provide information on the maximum lag used and rationale for it, for example in relation to the cycle widths of the observed oscillations on one hand and axonal conduction velocities on the other.*

Following the Reviewer's suggestion, we have now included more detail in the Methods' Section "The Time-Delay Embedded Hidden Markov Model":

"Therefore, besides the number of states (see Discussion), the important parameters of the model are the length of the window (i.e. the number of lags to be modelled by the state autocovariance matrices) and the number of PCA components. From a practical perspective, a trade-off between these two parameters will prescribe which frequencies in the data the TDE-HMM will be more sensitive to. That is, longer windows (more extended lags) and fewer principal components will incline the model to be more sensitive to the lower frequencies, whereas if we include more principal components and/or reduce the window we will be better able to capture high-frequency differences. In this paper, as mentioned earlier, we used a window of 60ms and 42x2 principal components. This window contains one cycle at exactly 16.6Hz, but note that this does not preclude the model from accessing slower frequencies (see Discussion)."

Minor Comment 4. *The authors have set the number of states in the HMM somewhat arbitrarily to 12. Why analyse 12 and proceed to ignore many of them when the 6 state division seems to convey all essential state information noted also for 12 states? Are the states hierarchically organized as in hierarchically modular networks? The authors note that the number of states essentially influences level of detail or resolution at which brain dynamics are viewed. While I agree with this notion, it would be good to provide statistical evidence that more than one stable state exists and is reliable in the first place and further use a measure of stability to quantify whether the chosen numbers of states yield valid/stable divisions into states (analogously to measuring the stability of graph module allocations).*

The question of how to best characterise hierarchies of states through separate HMM runs with different numbers of states is a key area of current investigation, and we are very glad that the Reviewer brings it up. It is our goal to provide a fully validated and complete solution to this question in the future, and, although for the purposes of this paper, we could easily reason through the (Riemannian) distances between the 12-states and the 6-states solution (similarly as we did in **Fig. SI-10a** when comparing HMM runs from different half-splits), this would not be completely satisfactory or as much useful in our opinion. More specifically, we would like to address this question through modelling (i.e. a hierarchical model containing different HMM solutions) instead of providing a simply (correlation-based) descriptive examination of these particular results. The development of this model is, arguably, a separate project in its own right.

This paper's purpose, besides introducing the method, is to have a closer examination to the so-called higher-order cognitive states. The reason because we did not elaborate on the other states is not because of their lack of interest, but because of the paper's focus being another one. That is, we preferred to go more in depth into the aspects of the data that these states represent, instead of being more general and less specific (as it would have been appropriate in more standard methods-like contribution). We mentioned this briefly in the Discussion:

"Of note, although this paper has paid closer examination to the four states represented in Fig. 2, these occupy on average only 30% of the total scanning time (see also Fig. SI-6)."

In response to Reviewer's Comment 2, we have shown the reliability of using 12 states. We agree this is a valid criterion to choose the number of states, as we now mentioned in the Section "Number of Brain States and State Reliability":

"Another important aspect (and a sensible way of guiding the choice of the number of states) is state reliability, that is, how robust are the states across, for example, half-splits of the data?"

In fact, this is partly the reason why we did not choose 16 states in the first place, being our original intention to select the highest number of states while still having a reliable solution. We preferred not to include these considerations in the paper because we were not fully exhaustive in searching the optimal number of states in this sense.

Why analyse 12 and proceed to ignore many of them when the 6 state division seems to convey all essential state information noted also for 12 states?

In this case, we disagree with the Reviewer's about the 6-states solution conveying all the relevant information within the 12-states solution. For example, the hemispheric separation of left vs. right temporal network states (see Fig. SI-10) is absent the 6-states solution. Other differences exist, also at the level of connectivity.

Minor Comment 5. *How do the power topographies of each state match with power-correlation networks reported in the original HMM paper (Baker et al. (2014)) by the same group? While the Baker et al. (2014) paper identified 8 states which corresponded to the different canonical RSNs to some extent, the power topographies (or the phase correlation networks) in this paper do not seem to systematically correspond to the RSNs per se. I do not see that they should, but it would be useful to the community to know whether they do or not. Are the differences simply driven by the power-vs.-raw difference (different observation models) for the HMM-based method?*

This is a good point. It is true that the states found here are not equivalent to those in Baker et al. (2014), and that they are also somewhat different from canonical fMRI RSNs. In order to address these two issues we have included two new analyses.

First, with regards to the equivalence to Baker et al. (2014)'s, we have now added a comparison between the Baker et al. method and the method proposed in this paper, both on the dataset used in this paper. This can be found in the new section in Results:

"To compare our results to previous work, we also ran the HMM on power envelopes computed from the data band-passed filtered between 1Hz and 40Hz (Baker et al., 2014), which defines HMM

states as having distinct patterns of power and power correlations in a single frequency band. We then computed the correlation between the power maps and the phase-coupling connectivity profiles (in either case using the multitaper after the HMM inference) between the two types of HMM. We paired the states between the two runs such that the correlations are maximal. **Fig. 7b** shows that some power maps are relatively well correlated between the two runs, and that the differences of functional connectivity between the two runs are in general larger than the differences in power. In summary, this analysis demonstrates that, while there are similarities between the two HMM approaches, there are also distinct characteristics to each of them.”

Second, with regards to differences to canonical fMRI RSNs we have now included a quantitative analysis in a new section in the Results. Please see the response to Comment 3 for details.

Minor Comment 6. *Comparison with the new Pascual-Marqui method could be presented quantitatively. Are all Pascual-Marqui connections also found by the Colclough method? For example, if Pascual-Marqui connections are considered a truth network, what is the sensitivity and specificity (or accuracy, TPR, FPR...) of the Colclough network?*

We fully agree that it will be important to quantitatively assess the relationship between the two methods in a wide range of contexts. However, we feel that it is beyond the scope of this paper (which already contains a large amount of methodological work), particularly since the two solutions are qualitatively very similar in this case (see **Fig. SI-12**).

Reviewer #3

Main Comment. *In order to make the paper significantly more powerful, the approach and the results should be better embedded in the context of other existing approaches and should be compared to them.*

We are grateful to the Reviewer for this suggestion and for the comprehensive relation of references, and we definitely agree that the paper would benefit from a better account of previous work. Although we are currently on the limit of references and number of words permitted by the journal, we have sought to cover as much as the suggested bibliography as possible (at the expense of some other references that were perhaps less relevant to the main points of the paper). We have also included further bibliographical suggestions from Reviewer #1.

In relation with EEG microstates, we have added the section "**Relation to EEG microstates**" with the following content:

“Our approach is not the first in proposing a segmentation of electrophysiological time series into a discrete set of states. For example, Rabinovich et al. (2015), among others, argue for the characterisation of brain dynamics as “a task-dependent sequential activations metastable states, that is, states where system variables reach and temporary hold stationary values” (see also Tognoli and Kelso (2014) for a general reference about metastability in the brain). A prominent related methodology is the EEG microstates framework (see e.g. van de Ville et al. (2010); Khanna et al. (2010)). One essential difference between the approach taken in this work and the EEG microstates is that we characterise HMM states in source-space. Most importantly, our states are specifically defined as periods in time where the data exhibits distinct spectral and cross-spectral properties. This allows us to identify states that correspond to networks of specific multivariate spectral

patterns, including coherence. By contrast, EEG microstates do not appear to exhibit distinct spectral properties. While other approaches applied to EEG data that do capture spectral differences have also been proposed, these are in sensor space, and so cannot capture the changes in phase-coupling between specific subnetworks of cortical regions that we find in this work (Koenig et al., 2001; Studer et al., 2006; Betzel et al., 2012)."

Also, we have added the new section "Biological underpinnings and functional labelling" in the Discussion, which reads as follows:

"In this paper, we have characterised large-scale brain states from MEG data, in terms of power and phase-coupling. In order to gain insight into the mechanistic underpinnings of these patterns and their dynamics, the descriptive methods presented here will need to be combined with biophysical modelling of large-scale networks (Woolrich and Stephan, 2013). With the exception of the higher-order cognitive functions, for which we have conducted a meta-analysis against existing literature, our state labelling is purely based on the anatomical location of power and functional connectivity. For example, we refer to the "visual" state as the state with wideband power and connectivity in occipital areas. However, when looking at the spectral characteristics of this state, we find that this activity is primarily occurring in the alpha band (see Fig. SI-4). Given the hypothesised inhibitory role of alpha (Jensen and Mazaheri, 2010; Klimesch, 2012), it is likely that this state is cognitively representing a reduction, rather than an increase, in visual activity. Some further connections to existing literature are plausible. For example, given its long-distance connections between the anterior temporal lobes and its low-frequency dominance, a relation between the anterior higher-order cognitive state and memory retrieval is very likely (Maguire and Mummery, 1999; Jensen and Tesche, 2002; Fell and Axmacher, 2011), yet cognitive control could also be involved (Cavanagh and Frank, 2014). Likewise, the posterior higher-order cognitive state could be related to attention and cross-modal processing (Behrmann et al., 2004). A separate question is about the mechanisms and causes of state switching, and whether these can be linked to avalanches of activity (Beggs and Plenz, 2003)."

Comment 1. *Only four out of twelve observed states are described and discussed in detail in the manuscript.*

Although it is true that much more information exists beyond these four states, we intentionally decided to focus on these in this paper. In particular, our original motivation for this work was to investigate the spectral and temporal properties of the DMN in MEG, and so our main interest is the two higher-cognitive states and their relation to the DMN. The other eight states are shown in Fig. SI-2.

What was the percentage of explained variance for these four states (relative to the total variance)?

We need to first clarify that explained variance is not an appropriate measure for this type of model. Explained variance is normally used in the context of a model that accounts for a percentage of the signal and leaves the rest as unexplained residuals (e.g. regression models or principal component analysis). In contrast, here we have a fully probabilistic model, where the description of each HMM state is a distinct probabilistic representation of the data, in this case in the form of a multivariate Gaussian distribution.

Therefore, a more appropriate measure is fractional occupancy, i.e. the percentage of time

where that state is the most probable description of the data. We have now referred to the fractional occupancy of the four states in the Discussion:

"Of note, although this paper has paid closer examination to the four states represented in Fig. 2, these occupy on average only 30% of the total scanning time."

Also, in relation to the fractional occupancy, we have now added the following paragraph in the Results (section "Higher-order cognitive states have distinct temporal characteristics"):

"A related point is the extent of state fractional occupancy variability across subjects, and whether this distribution is relatively flat or varies strongly across subjects. Fig. SI-6 reveals that the distribution is indeed not uniform, and that different subjects have different degrees of state representation, which might possibly relate to specific subject traits (Vidaurre et al., 2017b)."

Comment 2. *Power maps are illustrated relative to their temporal average. The scale would be of interest here, i.e. % change or dB in order to appreciate what 'low' and 'high' power means (in numbers). Furthermore, the thresholding criterion for the power maps is missing.*

Thanks - we have now indicated the exact scale in each map (Fig. 2, Fig. 4, Fig. SI-2 and Fig. SI-3). Also, as is now indicated in the Method section:

"For the wideband results (Fig. 2, Fig. SI-2 and Fig. SI-9), the power maps were thresholded such that only the 50% of voxels with the highest activation or deactivation are shown. For the frequency-specific results (Fig. 4 and Fig. SI-4), the threshold was set to 10%."

We have now also added actual statistical maps, reflecting which connections and activations are statistically significant (please see Question 2 from Reviewer #2 for details).

Comment 3. *Several other important parameters of the method are not justified and not described in detail. Specifically, why were 42 ROIs and 4 frequency profiles chosen?*

We now provide more detail about the choice of parcellation in the Methods' section "Data and preprocessing":

"Thirty-eight of these dipoles were obtained from a ICA decomposition on resting-state fMRI data from the Human Connectome Project, used previously to estimate large-scale static functional connectivity networks in MEG (Colclough et al., 2016); the other four parcels correspond to the anterior and posterior precuneus which we wanted to disambiguate from the PCC given the importance of this region in the resting state, and the left and right intraparietal sulci."

Note that a parcellation with approximately 40 ROIs is consistent with evidence that the effective dimensionality in MEG source space is approximately 64 (Taulu and Simola, 2006), and with the findings from using an adaptive parcellation approach (Farahibozorg et al, 2018). These references have now been included in the Supplementary References.

In relation to the choice of the number of frequency bands, we now provide some clarification in the section "Extracting spectral information" in Methods:

"... asking for four components, which we found to render stable decompositions while still

being reasonably frequency-specific. This choice of four components corresponds to the coarseness of classical frequency bands often used in the low frequency range we are studying (i.e. low gamma, beta, alpha, delta/theta)."

We have also added the following references in the Supplementary References:

S. Taulu and J. Simola, J. (2006). Spatiotemporal signal space separation method for rejecting nearby interference in MEG measurements. *Physics in Medicine and Biology* **51**, 1759–1768.

S.R. Farahibozorg, R.N. Henson and O. Hauk (2018). Adaptive cortical parcellations for source reconstructed EEG/MEG connectomes. *NeuroImage* **169**, 23–45.

Comment 4. Line 153 -> *It is said that frequency bands are split into the classical bands. However, delta/theta is defined as 0.5-10 Hz and alpha as 5-15 Hz. Besides the frequency overlap these are not the classical bands. I assume this is a typo.*

The reviewer makes a valid point. Following on from the previous comment, we have now clarified this in the section "Higher-order cognitive states have distinct spectral characteristics":

"The frequency modes were estimated following a data-driven approach (non-negative matrix factorisation, see Methods), which identified frequency modes that *approximately* correspond to classical frequency bands (although *overlap one another to a certain extent, bringing some data-driven flexibility*). For convenience, we labelled the data-driven modes using the *closest corresponding classical frequency bands, resulting in "delta/theta" (0.5-10Hz), "alpha" (5-15Hz), "beta" (15-30Hz) and "low gamma bands" (30-45Hz)*. It should however be kept in mind that the frequency modes are derived from the data, and so are not exactly the same as the classical frequency bands normally used."

Comment 5. Line 155 -> *Due to the relatively lower SNR in this band and considering the bias towards lower frequency activity resulting from running HMM on PCA components, it is more likely that gamma modes could not be observed given the used methods. What is crucially different from stating that there are no "state-specific differences" for these modes. This issue is addressed in the discussion; however, the wording should also be changed in the results section.*

We thank the Reviewer for pointing this. We have rephrased this as follows:

"Possibly due to the relatively low signal-to-noise ratio in higher frequency bands, *strong state-specific differences in the gamma band could not be observed with this approach*, and, therefore, we only show results for the delta/theta, alpha, and beta modes."

Comment 6. Line 317 -> *It seems that the authors down-sampled the data before filtering. If so, they do not mitigate the distortion due to aliasing. I assume it is a typo.*

We performed downsampling with an anti-aliasing filter. This has been now clarified in the methods as follows:

"MEG data were then downsampled to 250Hz *using an anti-aliasing filter, filtered out frequencies below 1Hz*, and source-reconstructed using LCMV beamforming "

Comment 7. Line 328 -> *By using the hidden Markov Model authors assumed that at each*

time point only one state is active. Moreover, the states are defined at group level and then the activation of a state at the subject level (Line 344). How do the authors justify that by using a 12-state model they are able both to explain the subject inter-variability and to guarantee that at each time point only one of this state is active?

The Reviewer raises a good point. It is worth noting that the HMM states are defined during the inference at the group level, that is they do not explain subject inter-variability excepting temporally (through the subject-specific state time courses). Therefore, each state is averaged across-subjects. Using a low number of components is, for example, standard in group-level ICA analyses; here, besides, the state distribution contains more information than an ICA component (i.e. the spectra). In any case, the model is more detailed than a static description, which is also standard in the field (Colclough et al., 2016). Also in relation to between-subjects differences, we now mention in the Discussion:

"While each state has its own subject-specific temporal characteristics (i.e. the state time courses), their spatial and spectral features are defined at the group level. However, note that if they are needed, e.g. to investigate their between-subject spatial and spectral differences, it is straightforward to re-compute subject-specific states features by combining the state time courses and data for each subject separately."

Comment 8. *Line 383 -> Authors applied PCA. It would be important to know the amount of variance explained by the decomposition per subject.*

We have now stated this in the Methods section:

"In this data set, this explains on average 60% of the variance (lowest and highest across subjects are, respectively, 55% and 66%)."

Comment 9. *Line 479 -> The authors should define "in numbers" what they assume saying "having more probability".*

We apologize for the confusion. In this case, we meant that we show *any* connection assigned to the larger Gaussian distribution. We have simplified this to:

"we only show the connections that belong to the Gaussian distribution representing the strongest connections".

Reviewers' comments:

Reviewer #1 (Remarks to the Author):

The authors have addressed all of my points and concerns with this revision. I think this paper will be of interest to a wide community, and will introduce a valuable new data-driven analysis methodology to neuroscientists studying the role of phase-coupled distributed networks in action, perception, and cognition.

Reviewer #2 (Remarks to the Author):

The authors have considered all prior review comments adequately and, in my opinion, significantly improved the manuscript. I see this study as an important opening both scientifically and methodologically, and thereby well suited for a top-level inter-disciplinary journal.

Reviewer #3 (Remarks to the Author):

Most of my concerns have been answered and corrections were done appropriately. However, my request to connect their method and results to other existing approaches has been answered somewhat superficially. It is true that one particularity of the proposed method is that it has been applied to MEG data transferred to source-space, while microstate analysis is performed on the sensor space. However, it is not evident that this is an "essential" difference as claimed by the authors. Since the contribution of power to the HMM state segmentation is 4 times higher than the contribution of coherence (as shown in the new Figure 7), it is not sure whether the analysis in source space is a major difference. Interestingly, studies that transformed the microstates into source space show very similar networks underlying the microstates as the ones underlying the HMM states (see for example Custo et al., Brain Connectivity 2017, where a similar distinction between an anterior and posterior DMN has been described). Also an earlier report of Pascual-Marqui et al., (arXiv, 2014) described the microstates as different parts of the DMN. In the paper of Milz et al. (Neuroimage 2017), the microstates were related to synchronization in the Alpha-band, which also dominate in the present work at least for the PCC and visual network.

The statement of the authors that microstates do not exhibit distinct spectral properties is not fully correct. As recently discussed in the review article by Michel & König (Neuroimage, 2018), EEG microstates are due to similar oscillations of the underlying sources with zero or minimal phase-lag, thus with high coherence. The crucial question is how much phase-lag the sources can exhibit so that still one stable topography is seen at the scalp level. In the approach presented by the authors zero-phase lag is excluded by the leakage correction, so the phase-coupling calculated by the coherence measure has certain phase-lags. I could not find a clear description of how much this phase-lag was in average. The authors refer to work by Fries and Engels concerning the importance of phase-locking for communication. However, the phase-lags in such coupled networks are very small. More information about the phase-lags and a more proper comparison of the networks derived from the HMM and the microstate approach would be appreciated.

Reviewer #3

Most of my concerns have been answered and corrections were done appropriately. However, my request to connect their method and results to other existing approaches has been answered somewhat superficially.

We thank Reviewer #3 for their comments. We fully agree that comparing the methods and results of our approach to microstate analyses is useful and appropriate in this context, and include this in our revision, as specified below. We have also included new information, in the form of an SI figure, with regards to the phase-lags identified by the HMM. Changes are highlighted in red.

Since the contribution of power to the HMM state segmentation is 4 times higher than the contribution of coherence (as shown in the new Figure 7), it is not sure whether the analysis in source space is a major difference.

We would like to clarify that having relatively small between-states differences in coherence compared to power (as examined post-hoc and according to the chosen metric) does not necessarily mean that coherence cannot have a significant impact on the HMM state segmentation. For example, as shown in **Fig. 7b**, the states inferred from the present time-delay embedded HMM have important differences from those obtained from an HMM that only uses power (Baker et al., 2014).

We can see how our previous description of these results may have led to the Reviewer's confusion on this point. We have now modified the wording in the corresponding section of the Results to make this clearer:

"... **However, this (referring to between-states differences) does not mean that phase coupling does not contribute to the inference. To demonstrate this, we** also ran the HMM on power envelopes ..."

We agree that substantial information about the states will be accessible in sensor space, but given the findings in **Fig. 7b**, we still expect there to be differences. A full comparison between analyses in sensor space and in (parcellated) source space would be instructive in future work. To make these points more clearly, we now have added the following to the Discussion:

"... **whereas the proposed model operates in source-space, microstates are estimated in sensor-space. Although HMM states can also be estimated in sensor space, source-reconstruction is however useful for noise removal and to better balance the contribution of deeper regions compared with more dominant (superficial) cortical areas.**"

Interestingly, studies that transformed the microstates into source space show very similar networks underlying the microstates as the ones underlying the HMM states (see for example Custo et al., Brain Connectivity 2017, where a similar distinction between an anterior and posterior DMN has been described). Also an earlier report of Pascual-Marqui et al., (arXiv, 2014) described the microstates as different parts of the DMN.

We agree that microstate analysis offers an important and complementary perspective on resting electrophysiological data. As such we have now included new text on microstates and their contribution to resting state network analysis, particularly in relation to the results we have in this

paper, as follows:

“Segmentation of EEG scalp maps into microstates is based on finding repeating distributions of power across multiple sensors, and therefore could be expected to capture interactions related to those that drive the HMM. Microstates have also offered new insights into the nature of resting state networks (Michel and König, 2018), including some evidence of a fragmentation of the DMN into anterior and posterior states (Pascual-Marqui et al., 2014; Custo et al., 2017). However, some fundamental differences exist. Most importantly, the HMM directly identifies states with distinct spectral and cross-spectral *profiles*, including coherence networks in distinct frequency bands and, potentially, at diverse phases (Maris et al., 2016). In contrast, while microstates can capture broadband spectral phenomena (Michel and König, 2018), their estimation (performed in sensor-space) is not based on spectral *profiles* (but see Milz et al., 2017), and assumes zero-lag (or 180°) phase differences. Also, whereas the proposed model operates in source-space, microstates are estimated in sensor-space. Although HMM states can also be estimated in sensor space, source-reconstruction is however useful for noise removal and to better balance the contribution of deeper regions compared with more dominant (superficial) cortical areas. In summary, it is through the use of the HMM that we have been able to reveal that the fragmentation of the DMN into anterior and posterior states is characterized by not only the presence of phase-locking networks, but also spectral power and phase-locking in distinct frequency bands.”

In the paper of Milz et al. (Neuroimage 2017), the microstates were related to synchronization in the Alpha-band, which also dominate in the present work at least for the PCC and visual network.

We thank the Reviewer for bringing up these references. However, a key property of the HMM segmentation is that the states not only exhibit a variety of distinct spectral properties at different frequency bands, but their estimation is crucially dependent on such spectral profiles. As a result, the approach is not only able to characterise networks in the alpha band as the Reviewer has indicated, but also in other frequency bands (e.g. sensorimotor in beta band, and the anterior higher-order cognitive in the delta/theta frequency band).

The statement of the authors that microstates do not exhibit distinct spectral properties is not fully correct. As recently discussed in the review article by Michel & König (Neuroimage, 2018), EEG microstates are due to similar oscillations of the underlying sources with zero or minimal phase-lag, thus with high coherence.

Although it is true that, as opposed to EEG microstates analysis, the proposed approach is specifically designed to capture spectral changes, the Reviewer is right in making this point. We have removed this sentence from this version of the paper, and as mentioned above, we now include the following text in the Discussion:

“... the HMM directly identifies states with distinct spectral and cross-spectral *profiles*, including coherence networks in distinct frequency bands and, potentially, at diverse phases (Maris et al., 2016). In contrast, while microstates can capture broadband spectral phenomena (Michel and König, 2018), their estimation (performed in sensor-space) is not based on spectral *profiles* (but see Milz et al., 2017)”

In the approach presented by the authors zero-phase lag is excluded by the leakage correction, so the phase-coupling calculated by the coherence measure has certain phase-lags. I could not find a clear description of how much this phase-lag was in average. The authors refer to work by Fries and Engels concerning the importance of phase-locking for communication. However, the phase-lags in such coupled networks are very small.

This is certainly a very good point. The Reviewer is correct that leakage correction does remove the zero-lag correlations *on average*. However, transient (close to) zero-lag synchronisation can still be identified. We now make this point in the Supplemental Discussion, along with a new supplemental Figure SI-11:

"A related issue is whether leakage correction, which makes the signals orthogonal across the entire time series, precludes completely zero-lag (or small lag) relationships, which are central to the theory of communication through coherence (Fries 2015). Importantly, leakage correction operates at the level of the entire time series, and so only removes zero-lag correlations *on average*. This means that it is still possible to have *transient* periods of zero- or small- lag synchronisation. Focusing on the anterior and posterior higher-order cognitive networks, **Fig. SI-11** illustrates this point by showing the phase at which different regions have a high coherence with the PCC (using a threshold of 0.05). Each dot thus represents a region with high coherence with PCC at the indicated frequency. Colours represent large-scale cortical areas. Importantly, due to the sign ambiguity issue, it is not possible to distinguish in-phase (0) from anti-phase coherence (π). If we assume that anti-phase actually represents in-phase relationships, this figure suggests that many of the (transient) phase-locking relationships are actually close to zero-lag."

More information about the phase-lags and a more proper comparison of the networks derived from the HMM and the microstate approach would be appreciated.

As indicated in our point-by-point response above, we have now included in the paper a discussion comparing and relating the results from the HMM with previous results obtained using microstate analyses. Bearing in mind the journal's space limitations, we have also included new information, in the form of an SI figure and discussion, about the specific phasic relations identified by the HMM.

REVIEWERS' COMMENTS:

Reviewer #3 (Remarks to the Author):

I thank the authors for their responses to my remaining comments. The changes made are appropriate and answered my questions. Of particular relevance is the added supplementary figure that shows that connections are close to zero phase lag, which confirms my assumption that the HMM states are reflecting similar characteristics than the microstates. I am looking forward to future work directly comparing the two approaches. I accept the paper in its current form without any request for revision.